# OneNet: Enhancing Time Series Forecasting Models under Concept Drift by Online Ensembling

**Yi-Fan Zhang**[1,2], **Qingsong Wen**[3,*] **Xue Wang**[3], **Weiqi Chen**[3], **Liang Sun**[3],
**Zhang Zhang**[1,2], **Liang Wang**[1,2], **Rong Jin**[3,†] **Tieniu Tan**[1,2]

[1]School of Artificial Intelligence, University of Chinese Academy of Sciences (UCAS)
[2] State Key Laboratory of Multimodal Artificial Intelligence Systems (MAIS), Institute of Automation
[3]Alibaba Group

## Abstract

Online updating of time series forecasting models aims to address the concept drifting problem by efficiently updating forecasting models based on streaming data. Many algorithms are designed for online time series forecasting, with some exploiting cross-variable dependency while others assume independence among variables. Given every data assumption has its own pros and cons in online time series modeling, we propose **On**line **e**nsembling **Net**work (OneNet). It dynamically updates and combines two models, with one focusing on modeling the dependency across the time dimension and the other on cross-variate dependency. Our method incorporates a reinforcement learning-based approach into the traditional online convex programming framework, allowing for the linear combination of the two models with dynamically adjusted weights. OneNet addresses the main shortcoming of classical online learning methods that tend to be slow in adapting to the concept drift. Empirical results show that OneNet reduces online forecasting error by more than **50**% compared to the State-Of-The-Art (SOTA) method. The code is available at https://github.com/yfzhang114/OneNet.

## 1 Introduction

In recent years, we have witnessed a significant increase in research efforts that apply deep learning to time series forecasting [Lim and Zohren, 2021, Wen et al., 2022]. Deep models have proven to perform exceptionally well not only in forecasting tasks, but also in representation learning, enabling the extraction of abstract representations that can be effectively transferred to downstream tasks such as classification and anomaly detection. However, existing studies have focused mainly on the batch learning setting, assuming that the entire training dataset is available beforehand, and the relationship between the input and output variables remains constant throughout the learning process. These approaches fall short in real-world applications where concepts are often not stable but change over time, known as concept drift [Tsymbal, 2004], where future data exhibit patterns different from those observed in the past. In such cases, re-training the model from scratch could be time-consuming. Therefore, it is desirable to train the deep forecaster online, incrementally updating the forecasting model with new samples to capture the changing dynamics in the environment.

The real world setting, termed online forecasting, poses challenges such as high noisy gradients compared to offline mini-batch training [Aljundi et al., 2019a], and continuous distribution shifts which can make the model learned from historical data less effective for the current prediction. While some studies have attempted to address the issues by designing advanced updating structures or learning objectives [Pham et al., 2023, You et al., 2021], they all rely on TCN backbones [Bai et al.,

---

*Corresponding author
†Work done at Alibaba Group, and now affiliated with Meta.

37th Conference on Neural Information Processing Systems (NeurIPS 2023).

Table 1: **A motivating example for online ensembling**, where the reported metric is MSE and the forecast horizon length is set as 48. Cells are colored on the basis of the MSE value, from low (red) to medium (white) to high (blue). Columns titled `cross-variable` refer to methods that focus on modeling cross-variable dependence, and columns titled `cross-time` refer to methods that only exploit the temporal dependence and assume independence among covariates. All methods use the same training and online adaptation strategy.

| Dataset | #Variables | Cross-Variable | | Cross-Time | | | Both | | | Ours |
|---|---|---|---|---|---|---|---|---|---|---|
| | | TCN | FSNet | Time-TCN | DLinear | PatchTST | CrossFormer | TS-Mixer | Fedformer | |
| ETTh2 | 7 | 0.910 | 0.846 | 1.307 | 6.910 | 2.716 | 5.772 | 3.060 | 1.620 | 0.609 |
| ETTm1 | 7 | 0.250 | 0.127 | 0.308 | 1.120 | 0.553 | 0.370 | 0.660 | 0.516 | 0.108 |
| WTH | 21 | 0.348 | 0.223 | 0.308 | 0.541 | 0.465 | 0.317 | 0.482 | 0.372 | 0.200 |
| ECL | 321 | 10.800 | 7.034 | 5.230 | 7.388 | 5.030 | 94.790 | 5.764 | 27.640 | 2.201 |

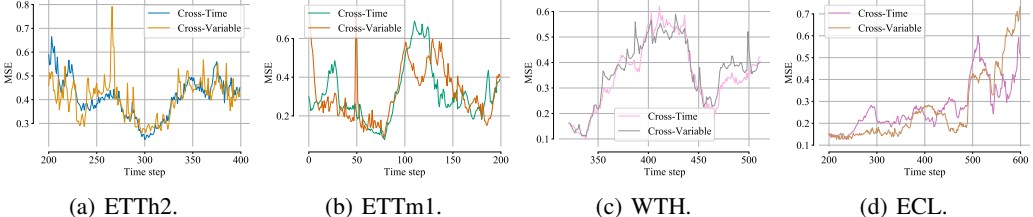

| (a) ETTh2. | (b) ETTm1. | (c) WTH. | (d) ECL. |

Figure 1: **A motivating example for online ensembling**, where the reported metric is MSE and forecast horizon length is set to 48 during online adaptation. Cross-Time refers to a TCN backbone that assumes independence among covariates and only models the temporal dependence, and cross-variable refers to a TCN backbone that takes into cross-variable dependence.

2018], which do not take advantage of more advanced network structures, such as transformer [Nie et al., 2023, Zhou et al., 2022b]. Our studies show that the current transformer-based model, PatchTST [Nie et al., 2023], without any advanced adaption method of online learning, performs better than the SOTA online adaptation model FSNet [Pham et al., 2023], particularly for the challenging ECL task (Table.1). Furthermore, we find that variable independence is crucial for the robustness of PatchTST. Specifically, PatchTST focuses on modeling temporal dependency (`cross-time dependency`) and predicting each variable independently. To validate the effectiveness of the variable independence assumption, we designed Time-TCN, which convolves only on the temporal dimension. Time-TCN is better than FSNet, a state-of-the-art approach for online forecasting, and achieves significant gains compared to the commonly used TCN structure that convolves on variable dimensions.

Although variable independence enhances model robustness, the `cross-variable dependency` is also critical for forecasting, i.e. for a specific variable, information from associated series in other variables may improve forecasting results. As shown in Table 1 for datasets ETTm1 and ETTh2, cross-time forecasters tend to yield lower performance for datasets with a small number of variables. Surprisingly, existing models that are designed to leverage both cross-variable and cross-time dependencies such as CrossFormer [Zhang and Yan, 2023] and TS-Mixer [Chen et al., 2023], tend to perform worse than a native TCN. To investigate this phenomenon, we visualized the MSE at different time steps during the entire online adaptation process for both a Cross-Time model (Time-TCN) and a Cross-Variable model (TCN) in Figure 1. We observe a large fluctuation in MSE over online adaption, indicating a significant concept drift over time. We also observe that neither of these two methods performs consistently better than the other, indicating that neither of the two data assumptions holds true for the entire time series. This is why relying on a single model like CrossFormer cannot solve this problem. Existing work depends on a simple model, but for online time series forecasting, data preferences for model bias will continuously change with online concept drifts. Therefore, we need a data-dependent strategy to continuously change the model selection policy. In other words, **online time series forecasting should go beyond parameter updating**.

In this paper, we address the limitation of a single model for online time series forecasting by introducing an ensemble of models that share different data biases. We then learn to dynamically combine the forecasts from individual models for better prediction. By allowing each model to be trained and online updated independently, we can take the best out of each online model; by dynamically

adjusting the combination of different models, we can take the best out of the entire model ensemble. We refer to our approach as Online Ensembling Network or **OneNet** for short. More concretely, OneNet maintains two online forecasting models, one focused on modeling temporal correlation and one focused on modeling cross-variable dependency. Each model is trained independently using the same set of training data. During testing, a reinforcement learning (RL) based approach is developed to dynamically adjust the weights used to combine the predictions of the two models. Compared to classical online learning methods such as Exponentiated Gradient Descent, our RL-based approach is more efficient in adapting to the changes/drifts in concepts, leading to better performance. The contributions of this paper are:

1. We introduce OneNet, a two-stream architecture for online time series forecasting that integrates the outputs of two models using online convex programming. OneNet leverages the robustness of the variable-independent model in handling concept drift, while also capturing the inter-dependencies among different variables to enhance forecasting accuracy. Furthermore, we propose an RL-based online learning approach to mitigate the limitations of traditional OCP algorithms and demonstrate its efficacy through empirical and theoretical analyses.

2. Our empirical studies with four datasets show that compared with state-of-the-art methods, OneNet reduces the average cumulative mean-squared errors (MSE) by $53.1\%$ and mean-absolute errors (MAE) by $34.5\%$. In particular, the performance gain on challenging dataset ECL is superior, where the MSE is reduced by $59.2\%$ and MAE is reduced by $63.0\%$.

3. We conducted comprehensive empirical studies to investigate how commonly used design choices for forecasting models, such as instance normalization, variable independence, seasonal-trend decomposition, and frequency domain augmentation, impact the model's robustness. In addition, we systematically compared the robustness of existing Transformer-based models, TCN-based models, and MLP-based models when faced with concept drift.

## 2 Preliminary and Related Work

**Concept drift.** Concepts in the real world are often dynamic and can change over time, which is especially true for scenarios like weather prediction and customer preferences. Because of unknown changes in the underlying data distribution, models learned from historical data may become inconsistent with new data, thus requiring regular updates to maintain accuracy. This phenomenon, known as concept drift [Tsymbal, 2004], adds complexity to the process of learning a model from data. In this paper, we focus on online learning for time series forecasting. Unlike most existing studies for online time series forecasting [Li et al., 2022, Qin et al., 2022, Pham et al., 2023] that only focus on how to online update their models, this work goes beyond parameter updating and introduces multiple models and a learnable ensembling weight, yielding rich and flexible hypothesis space. Due to the space limit, more related works about time series forecasting and reinforcement learning are left in the appendix.

**Online time series forecasting: streaming data.** Traditional time series forecasting tasks have a collection of multivariate time series with a look-back window $L$: $(\mathbf{x}_i)_{i=1}^{L}$, where each $\mathbf{x}_i$ is $M$-channel vector $\mathbf{x}_i = (x_i^j)_{j=1}^{M}$. Given a forecast horizon $H$, the target is to forecast $H$ future values $(\mathbf{x}_i)_{i=L+1}^{L+H}$. In real-world applications, the model builds on the historical data needs to forecast the future data, that is, given time offset $K' > L$, and $(\mathbf{x}_i)_{i=K'-L+1}^{K'}$, the model needs to forecast $(\mathbf{x})_{i=K'+1}^{K'+H}$. Online time series forecasting [Anava et al., 2013, Liu et al., 2016, Pham et al., 2023] is a widely used technique in real-world due to the sequential nature of the data and the frequent drift of concepts. In this approach, the learning process takes place over a sequence of rounds, where the model receives a look-back window and predicts the forecast window. The true values are then revealed to improve the model's performance in the next rounds. When we perform online adaptation, the model is retrained using the online data stream with the MSE loss over each channel: $\mathcal{L} = \frac{1}{M} \sum_{j=1}^{M} \| \hat{x}_{K'+1:K'+H}^j - x_{K'+1:K'+H}^j \|$.

**Variable-independent time series forecasting.** The traditional cross-variable strategy used in most structures takes the vector of all time series features as input and projects it into the embedding space to mix the information. On the contrary, PatchTST [Nie et al., 2023] adopts a variable-independent approach, where each input token only contains information from a single channel/variable. Our research demonstrates that variable independence is crucial for boosting model robustness under

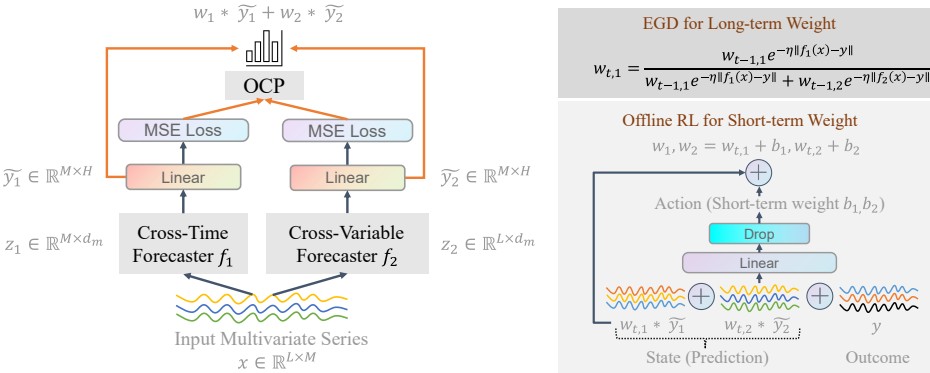

(a) **The overall OneNet architecture.**    (b) **OCP block.**

Figure 2: (a) OneNet processes multivariate data through cross-time and cross-variable branches, each responsible for capturing different aspects. The weights of these two branches are generated by the OCP block, and only the **black** arrows require execution during training. (b) The OCP block produces ensembling weights by utilizing both the long-term history of exponential gradient descent (EGD) and the short-term history of offline reinforcement learning (RL).

concept drift. For multivariate time series samples $(x_i^j)_{i=1}^{L}$, each channel $j$ is fed into the model independently, and the forecaster produces prediction results $(x_i^j)_{i=L+1}^{L+H}$ accordingly. As shown in Table 1, cross-variable methods tend to overfit when the dataset has a large number of variables, resulting in poor performance. This is evident in the poor performance of the SOTA online adaptation model FSNet [Pham et al., 2023] in the ECL dataset. However, models that lack cross-variable information perform worse on datasets with a small number of variables where cross-variable dependency can be essential. Although some existing work has attempted to incorporate both cross-variable interaction and temporal dependency into a single framework, our experiments show that these models are fragile under concept drift and perform no better than the proposed simple baseline, Time-TCN. To address this, we propose a novel approach that trains two separate branches, each focusing on modeling temporal and cross-variable dependencies, respectively. We then combine the results of these branches to achieve better forecasting performance under concept drift. We first introduce the OCP block for coherence.

## 3 OneNet: Ensemble Learning for Online Time Series Forecasting

We first examine online learning methods to dynamically adjust combination weights used by ensemble learning. We then present OneNet, an ensemble learning framework for online time series forecasting.

### 3.1 Learning the best expert by Online Convex Programming (OCP)

For notation clarity, here we denote $\mathbf{x} \in \mathbb{R}^{L \times M}$ as the historical data, $\mathbf{y} \in \mathbb{R}^{H \times M}$ as the forecast target. Our current method involves the integration of multiple complementary models. Therefore, how to better integrate model predictions in the online learning setting is an important issue. Exponentiated Gradient Descent (EGD) [Hill and Williamson, 2001] is a commonly used method. Specifically, the decision space $\triangle$ is a $d$-dimensional simplex, i.e. $\triangle = \{\mathbf{w}_t | w_{t,i} \geq 0 \text{ and } \| \mathbf{w}_t \|_1 = 1\}$, where $t$ is the time step indicator and we omit the subscript $t$ for simplicity when it's not confusing. Given the online data stream $\mathbf{x}$, its forecasting target $\mathbf{y}$, and $d$ forecasting experts with different parameters $\mathbf{f} = [\tilde{\mathbf{y}}_i = f_i(\mathbf{x})]_{i=1}^{d}$, the player's goal is to minimize the forecasting error as

$$\min_{\mathbf{w}} \mathcal{L}(\mathbf{w}) := \| \sum_{i=1}^{d} w_i f_i(\mathbf{x}) - \mathbf{y} \|^2; \quad s.t. \quad \mathbf{w} \in \triangle. \tag{1}$$

According to EGD, choosing $\mathbf{w}_1 = [w_{1,i} = 1/d]_{i=1}^d$ as the center point of the simplex and denoting $\ell_{t,i}$ as the loss of $f_i$ at time step $t$, the updating rule for each $w_i$ will be

$$w_{t+1,i} = \frac{w_{t,i} \exp(-\eta \parallel f_i(\mathbf{x}) - \mathbf{y} \parallel^2)}{Z_t} = \frac{w_{t,i} \exp(-\eta \ell_{t,i})}{Z_t} \tag{2}$$

where $Z_t = \sum_{i=1}^d w_{t,i} \exp(-\eta l_{t,i})$ is the normalizer, and the algorithm has a regret bound:

**Proposition 1.** *(**Online Convex Programming Bound**) For $T > 2\log(d)$, denote the regret for time step $t = 1, \ldots, T$ as $R(T)$, set $\eta = \sqrt{2\log(d)/T}$, the OCP updating policy have an **External regret** (See appendix B.1 for proof and analysis.)*

$$\sum_{t=1}^T \mathcal{L}(\mathbf{w}_t) - \inf_{\mathbf{u}} \sum_{t=1}^T \mathcal{L}(\mathbf{u}) \leq \sum_{t=1}^T \sum_{i=1}^d w_{t,i} \parallel f_i(\mathbf{x}) - \mathbf{y} \parallel^2 - \inf_{\mathbf{u}} \sum_{t=1}^T \mathcal{L}(\mathbf{u}) \leq \sqrt{2T\log(d)} \tag{3}$$

That is, the exponentially weighted average forecaster guarantees that the forecaster's cumulative expected loss is not much larger than the cumulative loss of the best decision. However, an exponentially weighted average forecaster is widely known to respond very slowly to drastic changes in the distribution [Cesa-Bianchi and Lugosi, 2006]. This phenomenon is sometimes referred to as the "**slow switch phenomenon**" in online learning literature, and is further illustrated in Figure 3 where the loss for $f_1$ is 0 for the first 50 trials and 1 for the next 50 trials. The performance of $f_2$ is the opposite. When the step size $\eta$ is small (e.g., $\eta = 0.01$), small changes are made to the weights and no clear adaptation takes place. When a large step size $\eta$ is applied (e.g., $\eta = 1$), we observe that the EGD algorithm quickly adapts to the environment change for the first 50 trials by increasing weight $w_1$ to almost 1 in the first few iterations. But it takes many iterations for the EGD algorithm to adapt to the change in the next 50 iterations, where $f_2$ works much better than $f_1$. We finally note that no matter how we adjust the step size $\eta$, the EGD algorithm has to suffer from the trade-off between speed of switching and overall good performance throughout the horizon.

Although few algorithms have been developed to address this issue in online learning [Stoltz and Lugosi, 2005, Cesa-Bianchi and Lugosi, 2003, Blum and Mansour, 2007, Foster and Vohra, 1998], the key idea is to find an activation function that maps the original policy $\mathbf{w}_t$ to a new one based on the recent loss of all experts. Despite the efforts, very limited successes have been achieved, either empirically or theoretically. In this work, we observe in our experiments that the combination weights $\mathbf{w}$ generated by the EGD algorithm are based on historical performance over a long period of time and thus cannot adapt quickly to transient environment changes. Hence, it is better to effectively incorporate both long-term historical information and more recent changes in the environment. A straightforward idea is to re-initialize the weight $\mathbf{w}$ per $K$ steps. We show that such a simple algorithm can achieve a tighter bound:

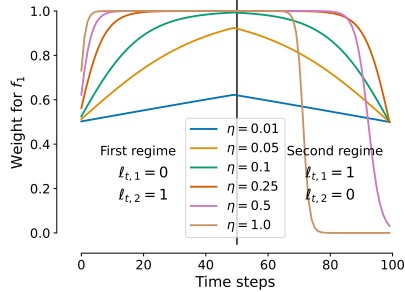

Figure 3: **The evolution of the weight assigned to** $f_1$ where the losses for forecasters vary across the first regime $[0, 50]$ and the second regime $[50, 100]$.

**Proposition 2.** *(**Informal**) Denote $I = [l, \cdots, r] \in [1, \cdots, T]$ as any period of time. We then have, the $K$-step re-initialize algorithm has a tighter regret bound compared to EGD at any small interval $I$, where $|I| < T^{\frac{3}{4}}$. (See appendix B.2 for proof.)*

Proposition 2 stresses that, by considering short-term information, we can attain lower regret in short time intervals. Such a simple strategy still struggles with the hyper-parameter choice of $K$. Besides, discarding long-term information makes the algorithm inferior to EGD for a long period of the online learning process. In this work, we address this challenge of online learning by exploiting offline reinforcement learning [Levine et al., 2020]. At first, we use EGD to maintain long-term weight $\mathbf{w}$. Besides, we introduce a different set of weights $\mathbf{b}$ that can better capture the recent performance of individual models. By combining $\mathbf{w}$ and $\mathbf{b}$, our approach can effectively incorporate both long-term historical information and more recent changes in the environment.

Specifically, we adopt the RvS [Emmons et al., 2022] framework, which formulates reinforcement learning through supervised learning, as shown in Figure 2(b). At time step $t$, our target is to learn

a short-term weight conditioned on the long-term weight $\mathbf{w}$ and experts' performances during a short period of history $I = [l, t]$. For simplicity and computation efficiency, we just let $l = t - 1$. The agent then chooses actions using a policy $\pi_{\theta_{rl}} \left( \mathbf{b}_t | \{ \{ w_{t,i} \tilde{y}_i \}_{i=1}^d \}_{t \in I}; \mathbf{y} \right)$ parameterized by $\theta_{rl}$. During training, we concatenate the product between each prediction and expert weight $(w_{t,i} * \tilde{\mathbf{y}}_i)$ with the outcome $\mathbf{y}$ as the conditional input. We follow RvS [Emmons et al., 2022] to implement the policy network as a two-layer MLPs $f_{rl} : \mathbb{R}^{H \times M \times (d+1)} \to \mathbb{R}^d$. Then the short-term weight and final ensembling weight will be:

$$\mathbf{b}_t = f_{rl} \left( w_{t,1} \tilde{\mathbf{y}}_1 \otimes \cdots \otimes w_{t,d} \tilde{\mathbf{y}}_d \otimes \mathbf{y} \right) \text{ and } \tilde{w}_{t,i} = (w_{t,i} + b_{t,i}) / \left( \sum_{i=1}^d (w_{t,i} + b_{t,i}) \right) \quad (4)$$

However, unlike in RvS, we cannot train the decision network through simple classification tasks since the ground truth target action is inaccessible. Instead, we propose to train the network by minimizing the forecasting error incurred by the new weight, that is, $\min_{\theta_{rl}} \mathcal{L}(\tilde{\mathbf{w}}) := \| \sum_{i=1}^d \tilde{w}_{t,i} f_i(\mathbf{x}) - \mathbf{y} \|^2$. During inference, as concept drift changes gradually, we use $\mathbf{w}_{t-1} + \mathbf{b}_{t-1}$ to generate the prediction and train the networks after the ground truth outcome is observed. We theoretically and empirically verify the effectiveness of the proposed OCP block in appendix B.4.

### 3.2 OneNet: utilizing the advantages of both structures

The model structure is shown in Figure 2(a) and we introduce the components as follows:

**Two-stream forecasters.** The input multivariate time series data is fed into two separate forecasters, a cross-time forecaster $f_1$ and a cross-variable forecaster $f_2$. Each forecaster contains an encoder and a prediction head. Assuming that the hidden dimension of the models are all $d_m$, the encoder of $f_1$ projects the input series to representation $z_1 \in \mathbb{R}^{M \times d_m}$, and the prediction head generates the final forecasting results: $\tilde{\mathbf{y}}_1 \in \mathbb{R}^{M \times H}$. For the cross-variable forecaster $f_2$, the encoder projects $\mathbf{x}$ to $\mathbf{z}_2 \in \mathbb{R}^{L \times d_m}$. Then, the representation of the last time step $\mathbf{z}_{2,L} \in \mathbb{R}^{d_m}$ is selected and fed into the prediction head to generate the final forecasting results $\tilde{\mathbf{y}}_2 \in \mathbb{R}^{M \times H}$. Compared to $f_1$, whose projection head has a parameter of $d_m \times H$, the projection head of $f_2$ has a parameter of $d_m \times M \times H$, which is heavier, especially when $M$ is large. Additionally, while $f_1$ ignores variable dependency, $f_2$ simply selects the representation of the last time step time series, ignoring temporal dependency. These two modules yield different but complementary inductive biases for forecasting tasks. *OCP block is then used for learning the best combination weights.* Specifically, we use EGD to update a weight $w_i$ for each forecaster and use offline-reinforcement learning to learn an additional short-term weight $b_i$, the final combination weight for one forecaster will be $w_i \leftarrow w_i + b_i$. Considering the difference between variables, we further construct different weights for each variable, namely, we will have $\mathbf{w} \in \mathbb{R}^{M \times 2}$ combination weights.

**Decoupled training strategy.** A straightforward training strategy for OneNet is to minimize $\mathcal{L}(w_1 * \tilde{\mathbf{y}}_1 + w_2 * \tilde{\mathbf{y}}_2, \mathbf{y})$ for both the OCP block and the two forecasters, where $w_i$ here denotes the weight with the additional bias term. However, the coupled training strategy has a fatal flaw: considering an extreme case where $f_1$ always performs much better than $f_2$, then $w_1$ will be close to 1 and $w_2 \to 0$. In this case, $\nabla_{\tilde{\mathbf{y}}_2} \mathcal{L}(w_1 * \tilde{\mathbf{y}}_1 + w_2 * \tilde{\mathbf{y}}_2, \mathbf{y}) \approx 0$, that is, $f_2$ is probably not trained for a long time. Under the context of concept drift, if retraining is not applied, as time goes on, the performance of $f_2$ will become much inferior. In this paper, therefore, we decouple the training process of the OCP block and the two forecasters. Specifically, the two forecasters is trained by $\mathcal{L}(\tilde{\mathbf{y}}_1, \mathbf{y}) + \mathcal{L}(\tilde{\mathbf{y}}_2, \mathbf{y})$ and the OCP block is trained by $\mathcal{L}(w_1 * \tilde{\mathbf{y}}_1 + w_2 * \tilde{\mathbf{y}}_2, \mathbf{y})$.

**Remark** Note that OneNet is complementary to advanced architectures for time series forecasting and online adaption methods under concept drift. A stronger backbone or better adaptation strategies/structure can both enhance performance.

## 4 Experiments

In this section, we will show that (1) the proposed OneNet attains superior forecasting performances with only a simple retraining strategy (reduce more than $50\%$ MSE compared to the previous SOTA model); (2) OneNet achieves faster and better convergence than other methods; (3) we conduct thorough ablation studies and analysis to reveal the importance of each of design choices of current advanced forecasting models. Finally, we introduce a variant of OneNet, called OneNet-, which has significantly fewer parameters but still outperforms the previous SOTA model by a large margin. Due to space limitations, some experimental settings and results are provided in the appendix.

Table 2: **MSE of various adaptation methods**. H: forecast horizon. OneNet-TCN is the mixture of TCN and Time-TCN, and OneNet is the mixture of FSNet and Time-FSNet.

| Method / H | ETTH2 | | | ETTm1 | | | WTH | | | ECL | | | |
|---|---|---|---|---|---|---|---|---|---|---|---|---|---|
| | 1 | 24 | 48 | 1 | 24 | 48 | 1 | 24 | 48 | 1 | 24 | 48 | Avg |
| Informer | 7.571 | 4.629 | 5.692 | 0.456 | 0.478 | 0.388 | 0.426 | 0.380 | 0.367 | - | - | - | 2.265 |
| OnlineTCN | 0.502 | 0.830 | 1.183 | 0.214 | 0.258 | 0.283 | 0.206 | 0.308 | 0.302 | 3.309 | 11.339 | 11.534 | 2.522 |
| TFCL | 0.557 | 0.846 | 1.208 | 0.087 | 0.211 | 0.236 | 0.177 | 0.301 | 0.323 | 2.732 | 12.094 | 12.110 | 2.574 |
| ER | 0.508 | 0.808 | 1.136 | 0.086 | 0.202 | 0.220 | 0.180 | 0.293 | 0.297 | 2.579 | 9.327 | 9.685 | 2.110 |
| MIR | 0.486 | 0.812 | 1.103 | 0.085 | 0.192 | 0.210 | 0.179 | 0.291 | 0.297 | 2.575 | 9.265 | 9.411 | 2.076 |
| DER++ | 0.508 | 0.828 | 1.157 | 0.083 | 0.196 | 0.208 | 0.174 | 0.287 | 0.294 | 2.657 | 8.996 | 9.009 | 2.033 |
| FSNet | 0.466 | 0.687 | 0.846 | 0.085 | 0.115 | 0.127 | 0.162 | 0.188 | 0.223 | 3.143 | 6.051 | 7.034 | 1.594 |
| Time-TCN | 0.491 | 0.779 | 1.307 | 0.093 | 0.281 | 0.308 | 0.158 | 0.311 | 0.308 | 4.060 | 5.260 | 5.230 | 1.549 |
| PatchTST | 0.362 | 1.622 | 2.716 | 0.083 | 0.427 | 0.553 | 0.162 | 0.372 | 0.465 | 2.022 | 4.325 | 5.030 | 1.512 |
| OneNet-TCN | 0.411 | 0.772 | 0.806 | 0.082 | 0.212 | 0.223 | 0.171 | 0.293 | 0.310 | 2.470 | 4.713 | 4.567 | 1.253 |
| OneNet | **0.380** | **0.532** | **0.609** | **0.082** | **0.098** | **0.108** | **0.156** | **0.175** | **0.200** | **2.351** | **2.074** | **2.201** | **0.747** |

## 4.1 Experimental setting

**Baselines of adaptation methods** We evaluate several baselines for our experiments, including methods for continual learning, time series forecasting, and online learning. Our first baseline is OnlineTCN [Zinkevich, 2003], which continuously trains the model without any specific strategy. The second baseline is Experience Replay (ER) [Chaudhry et al., 2019], where previous data is stored in a buffer and interleaved with newer samples during learning. Additionally, we consider three advanced variants of ER: TFCL [Aljundi et al., 2019b], which uses a task-boundary detection mechanism and a knowledge consolidation strategy; MIR [Aljundi et al., 2019a], which selects samples that cause the most forgetting; and DER++ [Buzzega et al., 2020], which incorporates a knowledge distillation strategy. It is worth noting that ER and its variants are strong baselines in the online setting, as we leverage mini-batches during training to reduce noise from single samples and achieve faster and better convergence. Finally, we compare our method to FSNet [Pham et al., 2023], which is the previous state-of-the-art online adaptation method. Considering different model structures, we compare the performance under concept drift of various structures, including TCN [Bai et al., 2018], Informer [Zhou et al., 2021], FEDformer [Zhou et al., 2022b], PatchTST [Nie et al., 2023], Dlinear [Zeng et al., 2023], Nlinear [Zeng et al., 2023], TS-Mixer [Chen et al., 2023].

**Strong ensembling baselines.** To verify the effectiveness of the proposed OCP block, we compare it with several ensembling baselines. Given the online inputs $\mathbf{x}$, predictions of each expert $\tilde{y}_1$, $\tilde{y}_2$, and the ground truth outcome $\mathbf{y}$, the final outcome $\tilde{\mathbf{y}}$ of different baselines will be as follows: (1) **Simple averaging**: we simply average the predictions of both experts to get the final prediction, i.e., $\tilde{\mathbf{y}} = \frac{1}{2}(\tilde{y}_1 + \tilde{y}_2)$. (2) **Gating mechanism** Liu et al. [2021]: we learn weights to the output of each forecaster, that is, $h = \mathbf{W}Concat(\tilde{y}_1, \tilde{y}_2) + \mathbf{b}; w_1, w_2 = softmax(h)$, and the final result is given by $\tilde{\mathbf{y}} = w_1 * \tilde{y}_1 + w_2 * \tilde{y}_2$. (3) **Mixture-of-experts** Jacobs et al. [1991], Shazeer et al. [2017]: we use the mixture of experts approach, where we first learn the weights $w_1$ and $w_2$ by applying a softmax function on a linear combination of the input, i.e., $h = \mathbf{W}\mathbf{x} + \mathbf{b}; w_1, w_2 = softmax(h)$, and then we obtain the final prediction by combining the predictions of both experts as $\tilde{\mathbf{y}} = w_1 * \tilde{y}_1 + w_2 * \tilde{y}_2$. (4) **Linear Regression (LR)**: we use a simple linear regression model to obtain the optimal weights, i.e., $[w_1, w_2] = (X^T X)^{-1} X^T y$, where $X = [\tilde{y}_1, \tilde{y}_2]$ and $y$ is the ground truth outcome. (5) **Exponentiated Gradient Descent (EGD)**: we use EGD to update the weights $w_1$ and $w_2$ separately without the additional bias. (6) **Reinforcement learning to learn the weight directly (RL-W)**: we use the bias term in the OCP block to update the weights based on the predictions of both experts and the ground truth outcome, i.e., the weight is only dependent on $\tilde{y}_1$, $\tilde{y}_2$, and $\mathbf{y}$, but not on the historical performance of each expert. For all baselines with trainable parameters, the training procedure is just the same as the proposed OCP block.

## 4.2 Online forecasting results

**Cumulative performance** Table.2 and Table.3 present the cumulative performance of different baselines in terms of mean-squared errors (MSE) and mean-absolute errors (MAE). In particular, Time-TCN and PatchTST exhibit strong performance and outperform the previous state-of-the-art model, FSNet [Pham et al., 2023]. The proposed OneNet-TCN (online ensembling of TCN and Time-

Table 3: **MAE of various adaptation methods**. H: forecast horizon. OneNet-TCN is the mixture of TCN and Time-TCN and OneNet is the mixture of FSNet and Time-FSNet.

| Method / H | ETTH2 | | | ETTm1 | | | WTH | | | ECL | | | Avg |
|---|---|---|---|---|---|---|---|---|---|---|---|---|---|
| | 1 | 24 | 48 | 1 | 24 | 48 | 1 | 24 | 48 | 1 | 24 | 48 | |
| Informer | 0.850 | 0.668 | 0.752 | 0.512 | 0.525 | 0.460 | 0.458 | 0.417 | 0.419 | - | - | - | |
| OnlineTCN | 0.436 | 0.547 | 0.589 | 0.085 | 0.381 | 0.403 | 0.276 | 0.367 | 0.362 | 0.635 | 1.196 | 1.235 | 0.543 |
| TFCL | 0.472 | 0.548 | 0.592 | 0.198 | 0.341 | 0.363 | 0.240 | 0.363 | 0.382 | 0.524 | 1.256 | 1.303 | 0.549 |
| ER | 0.376 | 0.543 | 0.571 | 0.197 | 0.333 | 0.351 | 0.244 | 0.356 | 0.363 | 0.506 | 1.057 | 1.074 | 0.498 |
| MIR | 0.410 | 0.541 | 0.565 | 0.197 | 0.325 | 0.342 | 0.244 | 0.355 | 0.361 | 0.504 | 1.066 | 1.079 | 0.499 |
| DER++ | 0.375 | 0.540 | 0.577 | 0.192 | 0.326 | 0.340 | 0.235 | 0.351 | 0.359 | 0.421 | 1.035 | 1.048 | 0.483 |
| FSNet | 0.368 | 0.467 | 0.515 | 0.191 | 0.249 | 0.263 | 0.216 | 0.276 | 0.301 | 0.472 | 0.997 | 1.061 | 0.448 |
| Time-TCN | 0.425 | 0.544 | 0.636 | 0.211 | 0.395 | 0.421 | 0.204 | 0.378 | 0.378 | 0.332 | 0.420 | 0.438 | 0.399 |
| PatchTST | 0.341 | 0.577 | 0.672 | 0.186 | 0.471 | 0.549 | 0.200 | 0.393 | 0.459 | 0.224 | 0.341 | 0.375 | 0.399 |
| OneNet-TCN | 0.374 | 0.511 | 0.543 | 0.191 | 0.319 | 0.371 | 0.221 | 0.345 | 0.356 | 0.411 | 0.513 | 0.534 | 0.391 |
| OneNet | **0.348** | **0.407** | **0.436** | **0.187** | **0.225** | **0.238** | **0.201** | **0.255** | **0.279** | **0.254** | **0.333** | **0.348** | **0.293** |

Table 4: **Ablation studies of ensembling methods (MSE results).**

| Method | | ETTH2 | | | ETTm1 | | | WTH | | | ECL | | | Avg |
|---|---|---|---|---|---|---|---|---|---|---|---|---|---|---|
| | | 1 | 24 | 48 | 1 | 24 | 48 | 1 | 24 | 48 | 1 | 24 | 48 | |
| Baseline | FSNet | 0.466 | 0.687 | 0.846 | 0.085 | 0.115 | 0.127 | 0.162 | 0.188 | 0.223 | 3.143 | 6.051 | 7.034 | 1.594 |
| Ensembling methods | Average | 0.381 | 0.607 | 0.595 | 0.088 | 0.105 | 0.111 | 0.154 | 0.176 | 0.197 | 2.458 | 2.833 | 3.309 | 0.918 |
| | Gating | 0.476 | 0.678 | 0.782 | 0.089 | 0.121 | 0.135 | 0.161 | 0.207 | 0.232 | 2.474 | 2.181 | 2.301 | 0.820 |
| | MOE | 0.488 | 0.565 | 1.238 | 0.084 | 0.107 | 0.133 | 0.155 | 0.179 | 0.213 | 3.312 | 3.086 | 2.497 | 1.005 |
| | LR | 0.741 | 0.634 | 0.589 | 0.153 | 0.107 | 0.113 | 0.229 | 0.179 | 0.198 | 4.376 | 2.235 | 2.478 | 1.003 |
| | EGD | 0.383 | 0.614 | 0.682 | 0.081 | 0.113 | 0.117 | 0.153 | 0.183 | 0.213 | 2.546 | 2.102 | 2.373 | 0.797 |
| | RL-W | 0.374 | 0.634 | 0.735 | 0.091 | 0.099 | 0.109 | 0.157 | 0.174 | 0.202 | 2.457 | 2.115 | 2.192 | 0.778 |
| Ours | OneNet- | 0.381 | 0.732 | 0.932 | 0.084 | 0.154 | 0.153 | 0.162 | 0.222 | 0.208 | 2.351 | 2.322 | 3.833 | 0.961 |
| | OneNet | 0.380 | 0.532 | 0.609 | 0.082 | 0.098 | 0.108 | 0.156 | 0.175 | 0.200 | 2.351 | 2.074 | 2.201 | 0.747 |

TCN) surpasses most of the competing baselines across various forecasting horizons. Interestingly, if the combined branches are stronger, for example, OneNet combined FSNet and Time-FSNet, achieving much better performance than OneNet-TCN. Namely, OneNet can integrate any advanced online forecasting methods or representation learning structures to enhance the robustness of the model. The average MSE and MAE of OneNet are significantly better than using either branch (FSNet or Time-TCN) alone, which underscores the significance of incorporating online ensembling.

**Comparison with strong ensembling baselines** is shown in Table 4. The two-branch framework greatly improves performance compared to FSNet with just simple ensembling methods such as averaging. The MOE approach that learns the weight from the input $\mathbf{x}$ performs poorly and is even inferior to simply averaging the prediction results. On the other hand, learning the weight from the prediction results in $\tilde{\mathbf{y}}_1$ and $\tilde{\mathbf{y}}_2$ (Gating) performing much better than MOE. This indicates that the combination weights should be dependent on the model prediction. However, formulating the learning problem as linear regression and using the closed-form solution is not a good idea due to the scarce nature of the online data stream and the high noise in the learned weight. EGD provides significant benefits compared to the averaging method, which highlights the importance of the cumulative historical performance of each expert. Additionally, we observe that RL-W achieves performance comparable or even better than EGD on some datasets. Therefore, we propose the OCP block that uses EGD to update the long-term weight and offline RL to learn the short-term weight. This design leads to superior performance compared to all the other baselines.

**Forecasting results are visualized in Figure 4**. Compared to baselines that struggle to adapt to new concepts and produce poor forecasting results, OneNet can successfully capture the patterns of time series. **More visualization results and convergence analysis** are presented in Appendix C.7.

## 4.3 Ablation studies and analysis

**The effect of instances normalization and seasonal-trend decomposition** is shown in Table. 5. The results show that removing the seasonal-trend decomposition component from PatchTST has limited

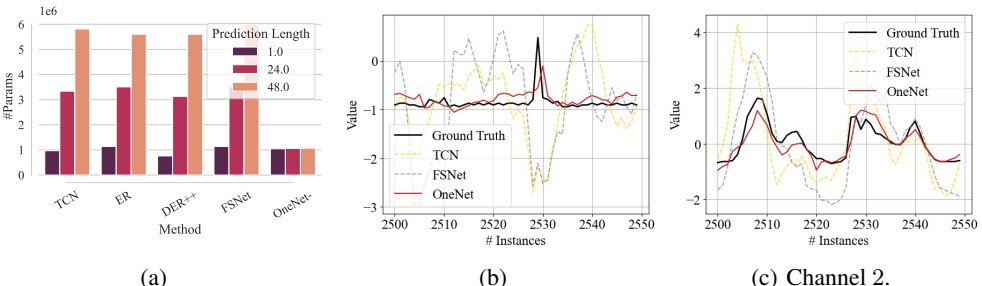

(a)             (b)             (c) Channel 2.

Figure 4: **Visualizing the model's prediction and parameters during online learning.** (a) Number of parameters for different models on the ECL dataset with different forecast horizons. We concentrate on a short 50-time step horizon, starting from $t = 2500$. (b), and (c) depict the model's prediction results for the first and second channels of the ECL dataset.

Table 5: **Ablation studies of the instances normalization (inv) and seasonal-trend decomposition (Decomp)** of non-adapted PatchTST and online adapted PatchTST, where the metric is MSE.

| Dataset | | | ETTH2 | | | ETTm1 | | | WTH | | | ECL | | | |
|---|---|---|---|---|---|---|---|---|---|---|---|---|---|---|---|
| Online | Inv | Decomp | 1 | 24 | 48 | 1 | 24 | 48 | 1 | 24 | 48 | 1 | 24 | 48 | Avg |
| ✓ | ✓ | ✗ | **0.360** | 1.625 | 2.670 | **0.083** | 0.436 | 0.555 | **0.161** | 0.370 | 0.464 | **1.988** | 4.345 | 5.060 | **1.510** |
| | ✗ | ✓ | 0.380 | 1.492 | 3.060 | 0.084 | 0.427 | **0.463** | 0.164 | 0.358 | **0.421** | 2.510 | 5.320 | 6.280 | 1.747 |
| | ✓ | ✓ | 0.362 | 1.622 | 2.716 | **0.083** | 0.427 | 0.553 | 0.162 | 0.372 | 0.465 | 2.022 | **4.325** | **5.030** | 1.512 |
| | ✗ | ✗ | 0.392 | **1.450** | **2.630** | 0.084 | **0.416** | 0.487 | 0.163 | **0.357** | 0.431 | 2.617 | 5.557 | 5.655 | 1.687 |
| ✗ | ✓ | ✗ | 0.397 | 2.090 | 3.156 | 0.084 | 0.448 | 0.553 | 0.161 | 0.372 | 0.467 | 2.000 | 4.398 | 5.100 | 1.602 |
| | ✗ | ✓ | 0.674 | 3.100 | 4.510 | 0.086 | 0.462 | 0.686 | 0.165 | 0.362 | 0.443 | 3.900 | 11.340 | 21.540 | 3.939 |
| | ✓ | ✓ | 0.427 | 2.090 | 3.290 | **0.083** | 0.433 | 0.570 | 0.163 | 0.375 | 0.467 | 2.030 | 4.395 | 5.101 | 1.619 |
| | ✗ | ✗ | 0.723 | 3.030 | 6.300 | 0.085 | 0.451 | 0.559 | 0.164 | 0.361 | 0.439 | 3.540 | 14.170 | 18.680 | 4.042 |

effect, regardless of whether the model is adapted online or not. Instances normalization is commonly used to mitigate the distribution shift between training and testing data, which is crucial for model robustness when online adaptation is impossible. However, when online adaptation is performed, the influence of instance normalization is reduced. Interestingly, our experiments reveal that *instance normalization impedes the model adaptation process in ETTH2, ETTm1, and WTH datasets when the forecast horizon is long (24 or 48)*. Thus, simply normalizing time series with zero mean and unit standard deviation may not be the optimal approach under concept drift. Ablation studies of the variable independence and frequency domain augmentation are detailed in the appendix.

**Delve deep into parameter-efficient online adaptation.** Although OneNet significantly reduces the forecasting error, it also increases the number of parameters and inference time due to its two-stream framework. We also design a variant of OneNet that may have slightly lower performance than OneNet, but with fewer parameters, making it more suitable for lightweight applications, denoted by OneNet-. Specifically, we ensemble PatchTST and Time-FSNet, which are both variable-independent. In this case, denote $\mathbf{z}_1, \mathbf{z}_2$ as the generated features for one variable from two branches, we concatenate the two features and feed them into the projection head, which further avoids the offline reinforcement learning block for ensembling weight learning and reduces the parameters. For example, in the ECL dataset, the hidden dimension FSNet [Pham et al., 2023] is 320, and the sequences have 321 channels. When the forecast horizon is 48, the projection head consists of just one linear layer with $320 \times 321 \times 48 = 4,930,560$ parameters. On the contrary, the concatenated features of OneNet- are always less than 1024 dimension, resulting in a final projection head with less than $1024 \times 48 = 49,152$ parameters. Figure 4(a) shows a detailed comparison of different methods on the ECL dataset. For small forecast horizons, all methods have a comparable number of parameters. As the forecast horizon increases, the number of parameters of existing adaptation methods increases rapidly. On the contrary, the number of parameters of OneNet- remains insensitive to the forecast horizon and is always less than all baselines. The performance of OneNet- is shown in Table 12, which is much better than FSNet but achieves fewer parameters.

See the appendix for the comparison of different forecasting models and more numerical results such as detailed ablation studies of different hyper-parameters and adaptation results under more settings.

# 5   Conclusion and Future Work

Through our investigation into the behavior of advanced forecasting models with concept drift, we discover that cross-time models exhibit greater robustness when the number of variables is large, but are inferior to models that can model variable dependency when the number of variables is small. In addition, this problem becomes more challenging due to the occurrence of concept drift, as the data preferences for both model biases are dynamically changing throughout the entire online forecasting process, making it difficult for a single model to overcome. To this end, we propose the OneNet model, which takes advantage of the strengths of both models through OCP. In addition, we propose to learn an additional short-term weight through offline reinforcement learning to mitigate the slow switch phenomenon commonly observed in traditional policy learning algorithms. Our extensive experiments demonstrate that OneNet is able to effectively deal with various types of concept drifts and outperforms previous methods in terms of forecasting performance.

We also discover that instance normalization enhances model robustness under concept drift, but can impede the model's ability to quickly adapt to new distributions in certain scenarios. This prompts further exploration of whether there exists a normalization technique that can mitigate distribution shifts while enabling rapid adaptation to changing concepts. In addition, although we design a lightened version of OneNet to address the problem of introducing additional parameters and inference time, there is potential for more efficient adaptation methods, such as utilizing prompts and efficient tuning methods from the NLP/CV community, to avoid retraining the full model.

## Acknowledgments and Disclosure of Funding

This work was supported by the National Key R&D Program of China (2022ZD0117901) and National Natural Science Foundation of China (Grant No. 62373355 and 62236010), and also supported by Alibaba Group through Alibaba Research Intern Program.

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

# OneNet: Enhancing Time Series Forecasting Models under Concept Drift by Online Ensembling

# —————Appendix—————

## Contents

# A Extended Related Work

**Concept Drift** Concepts in the real world are often dynamic and can change over time, which is especially true for scenarios like weather prediction and customer preferences. Because of unknown changes in the underlying data distribution, models learned from historical data may become inconsistent with new data, thus requiring regular updates to maintain accuracy. This phenomenon, known as concept drift [Tsymbal, 2004], adds complexity to the process of learning a model from data. Concept drift poses several challenging subproblems, ranging from fast learning under concept drift [Pham et al., 2023, Zhang et al., 2020, Gama et al., 2014], which involves adjusting the offline model with new observations to recognize recent patterns, to forecasting future data distributions [Li et al., 2022, Qin et al., 2022], which predicts the data distribution of the next time-step sequentially, enabling the model of the downstream learning task to be trained on the data sample from the predicted distribution. In this paper, we focus on the first problem, i.e. online learning for time series forecasting. Unlike most existing studies for online time series forecasting [Li et al., 2022, Qin et al., 2022, Pham et al., 2023] that only focus on how to online update their models, this work goes beyond parameter updating and introduces multiple models and a learnable ensembling weight, yielding rich and flexible hypothesis space.

**Test-Time adaptive methods** is similar to online time series forecasting but mainly focuses on domain generalization [Zhang et al., 2022c,a, Zhou et al., 2022a], domain adaptation [Zhang et al., 2023b] and other tasks [Liang et al., 2023]. These recently proposed to utilize target samples. Test-Time Training methods design proxy tasks during tests such as self-consistence [Zhang et al., 2021], rotation prediction [Sun et al., 2020] and need extra models; Test-Time Adaptation methods adjust model parameters based on unsupervised objectives such as entropy minimization [Wang et al., 2020] or update a prototype for each class [Iwasawa and Matsuo, 2021]. Domain adaptive method [Dubey et al., 2021] needs additional models to adapt to target domains. There are also some methods that do not need test-time tunning, for example, [Zhang et al., 2022b] introduces specific classifiers for different domains and adapts the voting weight for test samples dynamically and [Zhang et al., 2023a] apply non-parametric test-time adaptation.

**Time Series Modeling** Time series models have been developed for decades and are fundamental in various fields. While autoregressive models like ARIMA [Box and Pierce, 1970] were the first data-driven approaches, they struggle with nonlinearity and non-stationarity. Recurrent neural networks (RNNs) were designed to handle sequential data, with LSTM [Graves and Graves, 2012] and GRU [Chung et al., 2014] using gated structures to address gradient problems. Attention-based RNNs [Qin et al., 2017] use temporal attention to capture long-range dependencies, but are not parallelizable and struggle with long dependencies. Temporal convolutional networks [Sen et al., 2019] are efficient, but have limited reception fields and struggle with long-term dependencies. Recently, transformer-based [Vaswani et al., 2017, Wen et al., 2022] models have been renovated and applied in time series forecasting. Although a large body of work aims to make Transformer models more efficient and powerful [Zhou et al., 2022b, 2021, Nie et al., 2023], we are the first to evaluate the robustness of advanced forecasting models under concept drift, making them more adaptable to new distributions.

**Reinforcement learning and offline reinforcement learning.** Reinforcement learning is a mathematical framework for learning-based control, which allows us to automatically acquire policies that represent near-optimal behavioral skills to optimize user-defined reward functions [Hafner and Riedmiller, 2011, Silver et al., 2017]. The reward function specifies the objective of the agent, and the reinforcement learning algorithm determines the actions necessary to achieve it. However, the online learning paradigm of reinforcement learning is a major obstacle to its widespread adoption. The iterative process of collecting experience by interacting with the environment is expensive or dangerous in many settings, making offline reinforcement learning a more feasible alternative. Offline RL [Prudencio et al., 2023, Levine et al., 2020] learns exclusively from static datasets of previously collected interactions, enabling the extraction of policies from large and diverse training datasets. Effective offline RL algorithms have a much wider range of applications than online RL. Although there are many types of offline RL algorithms, such as those using value functions [Fujimoto et al., 2019], dynamics estimation [Kidambi et al., 2020], or uncertainty quantification [Agarwal et al., 2020], RvS [Emmons et al., 2022] has shown that simple conditioning with standard feedforward networks can achieve state-of-the-art results. In this work, we draw inspiration from RvS and learn an additional bias term for the OCP block for simplicity.

# B Proofs of Theoretical Statements

## B.1 Online Convex Programming Regret Bound

**Proposition 1.** *For $T > 2\log(d)$, denote the regret for time step $t = 1, \ldots, T$ as $R(T)$, set $\eta = \sqrt{2\log(d)/T}$, and the EGD update policy has regret.*

$$R(T) = \sum_{t=1}^{T} \mathcal{L}(\mathbf{w}_t) - \inf_{\mathbf{u}} \sum_{t=1}^{T} \mathcal{L}(\mathbf{u}) \leq \sum_{t=1}^{T} \sum_{i=1}^{d} w_{t,i} \parallel f_i(\mathbf{x}) - \mathbf{y} \parallel^2 - \sum_{t=1}^{T} \inf_{\mathbf{u}} \mathcal{L}(\mathbf{u}) \leq \sqrt{2T\log(d)} \tag{5}$$

*Proof.* The proof of Exponentiated Gradient Descent is well-studied [Ghai et al., 2020] and here we provide a simple Regret bound for both OCP.

Denote $\ell_{t,i} = \parallel f_i(\mathbf{x}) - \mathbf{y} \parallel^2$, recall that the normalizer for OCP-U is $Z_{t+1} = \sum_{i=1}^{d} w_{t,i} \exp(-\eta \ell_{t,i})$, then we have

$$\log \frac{Z_{t+1}}{Z_t} = \log \frac{\sum_{i=1}^{d} w_{t,i} \exp(-\eta \ell_{t,i})}{Z_t} = \log \sum_{i=1}^{d} p_{t,i} \exp(-\eta \ell_{t,i}), \tag{6}$$

where $p_{t,i} = w_{t,i}/Z_t \leq 1$. For clarity, we let $w_{t,i}$ be the unnormalized weight and $p_{t,i}$ be the normalized weight. Now, we assume $\eta \ell_{t,i} \in [0, 1]$. Although it is not guaranteed that $\ell_{t,i}$ will be small under concept shift, it is generally safe to assume that the concept will shift gradually and will not lead to a drastic change in the loss. As a result, the loss will not become arbitrarily large, and we can divide some large constant such that the loss is bounded in a small range. Based on the assumption, we can use the second Taylor expansion of $e^{-x} \leq 1 - x + x^2/2$ and the inequation $\log(1 - x) \leq -x$ for $x \in [0, 1]$. Then we have

$$\log \sum_{i=1}^{d} p_{t,i} \exp(-\eta \ell_{t,i}) \leq \log \left( 1 - \eta \sum_{i=1}^{d} p_{t,i} \ell_{t,i} + \frac{\eta^2}{2} \sum_{i=1}^{d} p_{t,i} \ell_{t,i}^2 \right) \tag{7}$$

$$\leq -\eta \sum_{i=1}^{d} p_{t,i} \ell_{t,i} + \frac{\eta^2}{2} \sum_{i=1}^{d} p_{t,i} \ell_{t,i}^2 \leq -\eta \sum_{i=1}^{d} p_{t,i} \ell_{t,i} + \frac{\eta^2}{2} \tag{8}$$

Note that $w_{t,i}$ is the unnormalized weight, and then $w_{1,i} = 1$ and $Z_1 = d$. Then we can get the lower bound and upper bound of $\log Z_{T+1}$:

$$\log Z_{T+1} = \sum_{t=1}^{T} \log \frac{Z_{t+1}}{Z_t} + \log(Z_1) \leq -\eta \sum_{t=1}^{T} \sum_{i=1}^{d} p_{t,i} \ell_{t,i} + \frac{T\eta^2}{2} + \log(d) \tag{9}$$

$$\log Z_{T+1} = \log \sum_{i=1}^{d} w_{t,i} \exp(-\eta \ell_{t,i}) \geq -\eta \sum_{i=1}^{d} \ell_{t,i} \tag{10}$$

Finally, recall that we set $\eta = \sqrt{2\log(d)/T}$, then we have

$$\eta \left( \sum_{t=1}^{T} \sum_{i=1}^{d} p_{t,i} \ell_{t,i} - \sum_{i=1}^{d} \ell_{t,i} \right) \leq \eta \left( \sum_{t=1}^{T} \sum_{i=1}^{d} p_{t,i} \ell_{t,i} - \inf_{\mathbf{u}} \sum_{t=1}^{T} \mathcal{L}(\mathbf{u}) \right) \leq \frac{T\eta^2}{2} + \log(d), \tag{11}$$

which completes our proof. $\qquad\qquad\qquad\qquad\qquad\qquad\qquad\qquad\qquad\qquad\qquad\square$

## B.2 Theoretical guarantee for the $K$-step re-initialize algorithm.

The proposed $K$-step re-initialize algorithm is detailed as follows: at the beginning of the algorithm, we choose $\mathbf{w}_1 = [w_{1,i} = 1/d]_{i=1}^{d}$ as the center point of the simplex and denote $\ell_{t,i}$ as the loss for $f_i$ at time step $t$, the updating rule for each $w_i$ will be $w_{t+1,i} = \frac{w_{t,i} \exp(-\eta[\partial \mathcal{L}_U(\mathbf{w}_t)]_i)}{Z_t} = \frac{w_{t,i} \exp(-\eta \parallel f_i(\mathbf{x}) - \mathbf{y} \parallel^2)}{Z_t} = \frac{w_{t,i} \exp(-\eta \ell_{t,i})}{Z_t}$, where $Z_t = \sum_{i=1}^{d} w_{t,i} \exp(-\eta l_{t,i})$ is the normalizer.

Different from the native EGD algorithm, we re-initialize the weight $\mathbf{w}_{K+1} = [w_{K+1,i} = 1/d]_{i=1}^d$ per $K$ time steps. We call each $K$ step one round. This simple strategy interrupts the influence of the historical information of length $K$ steps on the ensembling weights, which helps the model to quickly adapt to the upcoming environment.

**Proposition 2.** *For $T > 2\log(d)$, denote $I = [l, l+1, \cdots, r]$ as any period of time of length $r - l + 1$ where $l > 1$ and $r \leq T$. Denote the length of $I$ as a sublinear sequence of $T$, namely, $|I| = T^n$, where $0 \leq n \leq 1$. We choose $K = T^{\frac{2n}{3}}$. We then have, the $K$-step re-initialize algorithm has an regret bound $R(I) \leq \mathcal{O}(T^{2n/3})$ at any $I = [l, l+1, \cdots, r]$. Namely, for any small internal $n < \frac{3}{4}$, we have $R([l, l+1, \cdots, r]) < \mathcal{O}(T^{1/2})$*

*Proof.* We discuss regret in three cases:

- At first, according to Proposition 1, if all $K$ steps fall into $[l, l+1, \cdots, r]$, then we have $R(K) \leq 2\sqrt{K \ln d(d-1)}$. There exist $L/K$ rounds that are all contained in $[l, ..., r]$ and the regret of these rounds will be $\mathcal{O}(L/K * 2\sqrt{K})$.

- For the first round, we do not know when the weights are reinitialized ($l$ may or may not be a multiple of $K$) and the performance of the algorithm in historical time. let's think about the worst case, where regret will be less than $\mathcal{O}(K)$.

- For the last round, we know when the last round begins and the weights are reinitialized. However, some of the future time steps are not in $[l, l+1, \cdots, r]$ and we can only treat the case as the first round, which has a regret $\mathcal{O}(K)$.

Considering all cases, we have an internal regret that is bounded by

$$R(I) \leq \mathcal{O}(L/K \times \sqrt{K} + K) = \mathcal{O}(L/K\sqrt{K} + K) \tag{12}$$

When we choose an small internal length $L = T^n$ and $K = T^m$ where $m \leq n$. To minimize the upper bound, we should choose $m = \frac{2n}{3}$ and the bound will be $\mathcal{O}(T^{2n/3})$. Namely, for any small internal $n < \frac{3}{4}$, we have $R([l, l+1, \cdots, r]) < \mathcal{O}(T^{1/2})$, which is tighter than the regret bound of the EGD algorithm under the whole online sequence. Specifically, when we choose $L = T^{2/3}$ and $K = T^{4/9}$, we have $R([l, l+1, \cdots, r]) < \mathcal{O}(T^{4/9}) < \mathcal{O}(T^{1/2})$. When we choose $L = T^{1/4}$ and $K = T^{1/6}$, then $R([l, l+1, \cdots, r]) < \mathcal{O}(T^{1/6}) < \mathcal{O}(T^{1/2})$.

In other words, **the algorithm focuses on short-term information and leads to a better regret bound in any small time interval**. However, with increasing length of $I$, the bound of the simple algorithm will become worse. Consider an extreme case where $n = 1$, the bound will be $R([l, l+1, \cdots, r]) < \mathcal{O}(T^{2/3})$, which is inferior to the native EGD algorithm.

$\square$

### B.3 Necessary definitions and assumptions for evaluating model adaptation speed to environment changes.

To complete the proofs, we begin by introducing some necessary definitions and assumptions. Given the online data stream $\mathbf{x}_t$ and its forecasting target $\mathbf{y}_t$ at time $t$. Given $d$ forecasting experts with different parameters $\mathbf{f}_t = \{f_{t,i}\}$, denote $\ell$ as a nonnegative loss function and $\ell_{t,i} := \ell(f_{t,i}(\mathbf{x}_t), \mathbf{y}_t)$ as the loss incurred by $f_{t,i}$ at time t, we define the following notions.

**Definition 1.** *(Weighted average forecaster).* *A weighted average forecaster makes predictions by*

$$\tilde{\mathbf{y}}_t = \frac{\sum_{i=1}^d w_{t-1,i} f_{t,i}}{\sum_{i=1}^d w_{t-1,i}}, \tag{13}$$

*where $w_{t,i}$ is the weight for expert $f_i$ at time $t$ and $f_{t,i}$ is the prediction of $f_i$ at time $t$.*

**Definition 2.** *(**Cumulative regret and instantaneous regret**). For expert $f_i$, the cumulative regret (or simply regret) on the $T$ steps is defined by*

$$R_{T,i} = \sum_{t=1}^{T} r_{t,i} = \sum_{t=1}^{T} \left( \ell(\tilde{\mathbf{y}}_t, \mathbf{y}_t) - \ell_{t,i} \right) = \hat{L}_T - L_{T,i} \tag{14}$$

*where $r_{t,i}$ is the instantaneous regret of the expert $f_i$ at time $t$, which is the regret that the forecaster feels of not having listened to the advice of the expert $f_i$ right after the $t$th outcome that has been revealed. $\hat{L}_T = \sum_{t=1}^{T} \ell(\tilde{\mathbf{y}}_t, \mathbf{y}_t)$ is the cumulative loss of the forecaster and $L_{T,i}$ is the cumulative loss of the expert $f_i$.*

**Definition 3.** *(**Potential function**). We can interpret the weighted average forecaster in an interesting way which allows us to analyze the theoretical properties easier. To do this, we denote $\mathbf{r}_t = (r_{t,1}, \ldots, r_{t,d}) \in \mathbb{R}^d$ as the instantaneous regret vector, and $\mathbf{R}_T = \sum_{t=1}^{T} \mathbf{r}_t$ is the corresponding regret vector. Now, we can introduce the potential function $\Phi : \mathbb{R}^d \to \mathbb{R}$ of the form*

$$\Phi(\mathbf{u}) = \psi \left( \sum_{i=1}^{d} \phi(u_i) \right) \tag{15}$$

*where $\phi : \mathbb{R} \to \mathbb{R}$ is any nonnegative, increasing, and twice differentiable function, and $\psi : \mathbb{R} \to \mathbb{R}$ is nonnegative, strictly increasing, concave, and twice differentiable auxiliary function. With the notion of potential function, the prediction $\tilde{\mathbf{y}}_t$ will be*

$$\tilde{\mathbf{y}}_t = \frac{\sum_{i=1}^{d} \nabla \Phi(\mathbf{R}_{t-1})_i f_{t,i}}{\sum_{i=1}^{d} \nabla \Phi(\mathbf{R}_{t-1})_i}, \tag{16}$$

*where $\nabla \Phi(\mathbf{R}_{t-1})_i = \partial \Phi(\mathbf{R}_{t-1})/\partial R_{t-1,i}$. It is easy to prove that the exponentially weighted average forecaster used in Eq.(2) is based on the potential $\Phi_\eta(\mathbf{u}) = \frac{1}{\eta} \ln \left( \sum_{i=1}^{d} e^{\eta u_i} \right)$.*

**Theorem 1.** *(**Blackwell condition, Lemma 2.1. in [Cesa-Bianchi and Lugosi, 2006]**.) If the loss function $\ell$ is convex in its first argument and we use $\mathbf{x}_1 \cdot \mathbf{x}_2$ denote the inner product of two vectors, then*

$$\sup_{\mathbf{y}_t} \mathbf{r}_t \cdot \nabla \Phi(\mathbf{R}_{t-1}) \leq 0 \tag{17}$$

The following theorem is applicable to any forecaster that satisfies the Blackwell condition, not limited to weighted average forecasters. Nevertheless, this theorem will lead to several interesting bounds for various variations of the weighted average forecaster.

**Theorem 2.** *(Theorem2.1 in [Cesa-Bianchi and Lugosi, 2006].) Assume that a forecaster satisfies the Blackwell condition for a potential $\Phi$, then for all $i = 1, \cdots ,$*

$$\Phi(\mathbf{R}_T) \leq \Phi(0) + \frac{1}{2} \sum_{t=1}^{T} C(\mathbf{r}_t), \tag{18}$$

*where*

$$C(\mathbf{r}_t) = \sup_{\mathbf{u} \in \mathbb{R}^d} \psi' \left( \sum_{i=1}^{d} \phi(u_i) \right) \sum_{i=1}^{d} \phi''(u_i) r_{t,i}^2. \tag{19}$$

## B.4 Existing theoretical intuition and empirical comparison to the proposed OCP block.

With the help of the two theorems in Section B.3, we now recall that the **Internal Regret** $R_{in}(t, \mathbf{w})$ [Blum and Mansour, 2007] that measures forecaster's expected regret of having taken an action $\mathbf{w}$ at step $t$:

$$R_{in}(T, \mathbf{w}) = \max_{i,j=1,\ldots,d} \sum_{t=1}^{T} r_{t,(i,j)} = \max_{i,j=1,\ldots,d} \sum_{t=1}^{T} w_{t,i} \left( \ell_{t,i} - \ell_{t,j} \right). \tag{20}$$

While Proposition 1 ensures that a small external regret can be achieved, ensuring a small internal regret is a more challenging task. This is because any algorithm with a small internal regret also

has small external regret but the opposite is not true, as demonstrated in [Stoltz and Lugosi, 2005]. The key question now is whether it is possible to define a policy $\mathbf{w}$ that attains small (i.e., sublinear in $T$) internal regret. For simplicity, we use $R$ as internal regret in this subsection. To develop a forecasting strategy that can guarantee a small internal regret. We define the exponential potential function $\Phi : \mathbb{R}^M \to \mathbb{R}$ with $\eta > 0$ by

$$\Phi(\mathbf{u}) = \frac{1}{\eta} \ln \left( \sum_{i=1}^{M} e^{\eta u_i} \right), \tag{21}$$

where $M = d(d-1)$. Here, we denote $\mathbf{r}_t = (r_{t,(1,1)}, r_{t,(1,2)}, \ldots, r_{t,(d,d-1)}) \in \mathbb{R}^{d(d-1)}$ as the instantaneous regret vector and $\mathbf{R}_T = \sum_{t=1}^{T} \mathbf{r}_t$ is the corresponding regret vector. Then, any forecaster satisfying Blackwell's condition will have a bounded internal regret (Corollary 8 in [Cesa-Bianchi and Lugosi, 2003]) by choosing a proper parameter $\eta$:

$$\max_{i,j} R_{t,(i,j)} \le 2\sqrt{t \ln d(d-1)} \tag{22}$$

With the help of the two theorems in Section B.3, we now recall that the **Internal Regret** $R_{in}(t, \mathbf{w})$ [Blum and Mansour, 2007] that measures forecaster's expected regret of having taken an action $\mathbf{w}$ at step $t$:

$$R_{in}(t, \mathbf{w}) = \max_{i,j=1,\ldots,d} \sum_{t=1}^{T} r_{t,(i,j)} = \max_{i,j=1,\ldots,d} \sum_{t=1}^{T} w_{t,i} \left( \ell_{t,i} - \ell_{t,j} \right). \tag{23}$$

For simplicity, we use $R$ as internal regret in this subsection. To conduct a forecasting strategy that can guarantee a small internal regret. We define the exponential potential function $\Phi : \mathbb{R}^M \to \mathbb{R}$ with $\eta > 0$ by

$$\Phi(\mathbf{u}) = \frac{1}{\eta} \ln \left( \sum_{i=1}^{M} e^{\eta u_i} \right), \tag{24}$$

where $M = d(d-1)$. Here, we denote $\mathbf{r}_t = (r_{t,(1,1)}, r_{t,(1,2)}, \ldots, r_{t,(d,d-1)}) \in \mathbb{R}^{d(d-1)}$ as the instantaneous regret vector and $\mathbf{R}_T = \sum_{t=1}^{T} \mathbf{r}_t$ is the corresponding regret vector. Then, any forecaster satisfying Blackwell's condition will have a bounded internal regret (Corollary 8 in [Cesa-Bianchi and Lugosi, 2003]) by choosing a proper parameter $\eta$:

$$\max_{i,j} R_{t,(i,j)} \le 2\sqrt{t \ln d(d-1)} \tag{25}$$

Now our target is to find a new policy that makes the forecaster satisfy the Blackwell condition.

$$
\begin{aligned}
\nabla \Phi(\mathbf{R}_{t-1}) \cdot \mathbf{r}_t &= \sum_{i,j=1}^{d} \nabla_{(i,j)} \Phi(\mathbf{R}_{t-1}) w_{t,i} \left( \ell_{t,i} - \ell_{t,j} \right) \\
&= \sum_{i=1}^{d} \sum_{j=1}^{d} \nabla_{(i,j)} \Phi(\mathbf{R}_{t-1}) w_{t,i} \ell_{t,i} - \sum_{i=1}^{d} \sum_{j=1}^{d} \nabla_{(i,j)} \Phi(\mathbf{R}_{t-1}) w_{t,i} \ell_{t,j} \\
&= \sum_{i=1}^{d} \sum_{j=1}^{d} \nabla_{(i,j)} \Phi(\mathbf{R}_{t-1}) w_{t,i} \ell_{t,i} - \sum_{j=1}^{d} \sum_{i=1}^{d} \nabla_{(j,i)} \Phi(\mathbf{R}_{t-1}) w_{t,j} \ell_{t,i} \\
&= \sum_{i=1}^{d} \ell_{t,i} \left( \sum_{j=1}^{d} \nabla_{(i,j)} \Phi(\mathbf{R}_{t-1}) w_{t,i} - \sum_{k=1}^{d} \nabla_{(k,i)} \Phi(\mathbf{R}_{t-1}) w_{t,k} \right)
\end{aligned} \tag{26}
$$

To ensure that this value is negative or zero, it is sufficient to demand that.

$$\sum_{i=1}^{d} \ell_{t,i} \left( \sum_{j=1}^{d} \nabla_{(i,j)} \Phi(\mathbf{R}_{t-1}) w_{t,i} - \sum_{k=1}^{d} \nabla_{(k,i)} \Phi(\mathbf{R}_{t-1}) w_{t,k} \right) = 0, \forall i = 1, \cdots, d \tag{27}$$

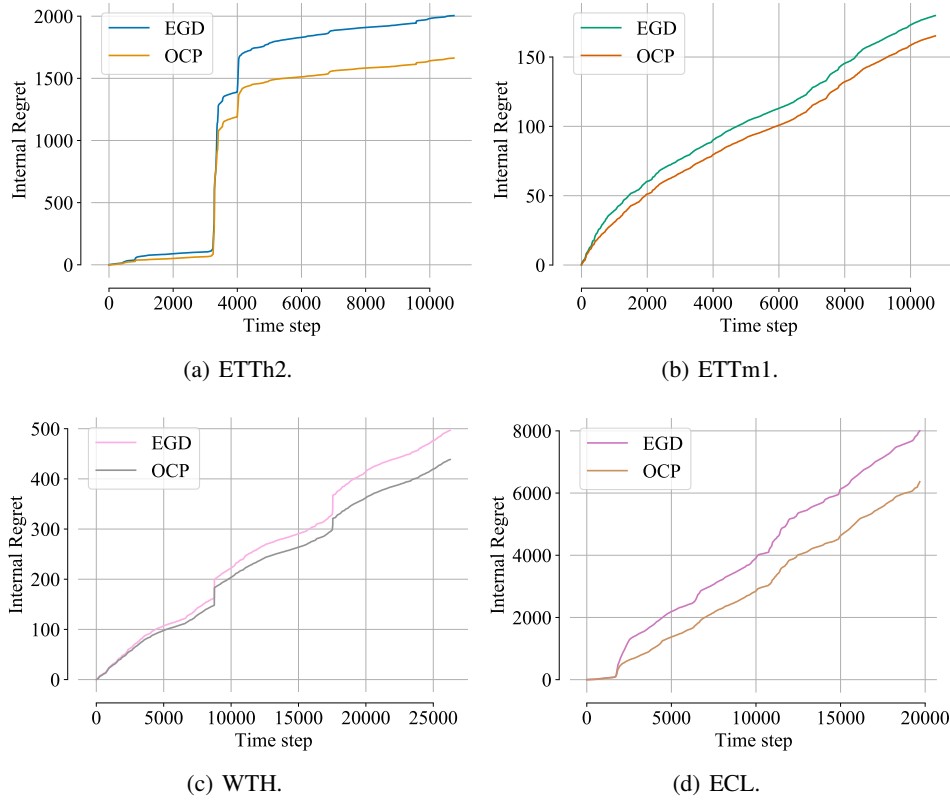

(a) ETTh2.

(b) ETTm1.

(c) WTH.

(d) ECL.

Figure 5: Empirical verification of the proposed OCP block can significantly reduce the internal regret compared to vanilla EGD, where the forecasting window $H = 48$.

That is, we need to find a new policy vector $\mathbf{w}_t$ that satisfies $\mathbf{w}_t^T A = 0$, where $A = \begin{cases} -\nabla_{k,i}\Phi(\mathbf{R}_{t-1}) & \text{if } i \neq k, \\ \sum_{j\neq i}\nabla_{k,j}\Phi(\mathbf{R}_{t-1}) & \text{Otherwise.} \end{cases}$ is an $d \times d$ matrix. However, determining the existence of a new policy vector and efficiently calculating its values can be challenging. Even if we assume the new vector exists, the time complexity of calculating the new policy by the Gaussian elimination method[Foster and Vohra, 1999] is $O(d^3)$, which is expensive, particularly for datasets with a large number of variables such as the ECL dataset with 321 variables and 321*2 policies. To address this issue, we propose the OCP block which utilizes an additional offline reinforcement learning block $f_{rl}$ with parameter $\theta_{rl}$ to learn a bias vector $\mathbf{b}_t$ for the original policy $\mathbf{w}_t$. The new policy vector is then defined as $\tilde{\mathbf{w}}_t = \mathbf{w}_t + \mathbf{b}_t$. The learned bias pushes the predicted outcomes closer to the ground truth values, that is, we minimize $\min_{\theta_{rl}} \mathcal{L}(\tilde{\mathbf{w}}) := \parallel \sum_{i=1}^d \tilde{w}_i f_i(\mathbf{x}) - \mathbf{y} \parallel^2; \quad s.t. \quad \tilde{\mathbf{w}} \in \triangle$ to train $\theta_{rl}$. We measure the internal regret $\max_{i,j=1,\dots,d} w_{t,i}(\ell_{t,i} - \ell_{t,j})$ at each time step empirically. As shown in Figure 5, the proposed method significantly reduces internal regret without the need for constructing and computing a large matrix.

## C Additional Experimental Results

### C.1 Datasets

We investigate a diverse set of datasets for time series forecasting. ETT [Zhou et al., 2021][3] logs the target variable of the "oil temperature" and six features of the power load over a two-year period. We also analyze the hourly recorded observations of ETTh2 and the 15-minute intervals of ETTm1 benchmarks. Additionally, we study ECL[4] (Electricity Consuming Load), which gathers electricity

---

[3]https://github.com/zhouhaoyi/ETDataset

[4]https://archive.ics.uci.edu/ml/datasets/ElectricityLoadDiagrams20112014

consumption data from 321 clients between 2012 and 2014. The Weather (WTH)[5] dataset contains hourly records of 11 climate features from almost 1,600 locations across the United States.

## C.2 Implementation Details

For all benchmarks, we set the look-back window length at 60 and vary the forecast horizon from $H = 1, 24, 48$. We split the data into two phases: warm-up and online training, with a ratio of 25:75. We follow the optimization details outlined in [Zhou et al., 2021] and utilize the AdamW optimizer [Loshchilov and Hutter, 2017] to minimize the mean squared error (MSE) loss. To ensure a fair comparison, we set the epoch and batch sizes to one, which is consistent with the online learning setting. We make sure that all baseline models based on the TCN backbone use the same total memory budget as FSNet, which includes three times the network sizes: one working model and two exponential moving averages (EMAs) of its gradient. For ER, MIR, and DER++, we allocate an episodic memory to store previous samples to meet this budget. For transformer backbones, we find that a large number of parameters do not benefit the generalization results and always select the hyperparameters such that the number of parameters for transformer baselines is fewer than that for FSNet. In the warm-up phase, we calculate the mean and standard deviation to normalize the online training samples and perform hyperparameter cross-validation. For different structures, we use the optimal hyperparameters that are reported in the corresponding paper.

**License.** All the assets (i.e., datasets and the codes for baselines) we use include an MIT license containing a copyright notice and this permission notice shall be included in all copies or substantial portions of the software.

**Environment.** We conduct all the experiments on a machine with an Intel R Xeon (R) Platinum 8163 CPU @ 2.50GHZ, 32G RAM, and four Tesla-V100 (32G) instances. All experiments are repeated 3 times with different seeds.

**Metrics** Because learning occurs over a sequence of rounds. At each round, the model receives a look-back window and predicts the forecast window. All models are commonly evaluated by their accumulated mean-squared errors (MSE) and mean-absolute errors (MAE), namely the model is evaluated based on its accumulated errors over the entire learning process.

## C.3 Baseline details

We present a brief overview of the baselines employed in our experiments.

First, OnlineTCN adopts a conventional TCN backbone [Zinkevich, 2003] consisting of ten hidden layers, each layer containing two stacks of residual convolution filters.

Secondly, ER [Chaudhry et al., 2019] expands on the OnlineTCN baseline by adding an episodic memory that stores previous samples and interleaves them during the learning process with newer ones.

Third, MIR [Aljundi et al., 2019a] replaces the random sampling technique of ER with its MIR sampling approach, which selects the samples in the memory that cause the highest forgetting and applies ER to them.

Fourthly, DER++ [Buzzega et al., 2020] enhances the standard ER method by incorporating a knowledge distillation loss on the previous logits.

Finally, TFCL [Aljundi et al., 2019b] is a task-free, online continual learning method that starts with an ER process and includes a task-free MAS-styled [Aljundi et al., 2018] regularization.

All the ER-based techniques utilize a reservoir sampling buffer, which is identical to that used in [Pham et al., 2023].

## C.4 Hyper-parameters

For the hyper-parameters of FSNet and the baselines mentioned in Section C.3, we follow the setting in [Pham et al., 2023]. Besides, we cross-validate the hyper-parameters on the ETTh2 dataset and use them for the remaining ones. In particular, we use the following configuration:

---

[5]https://www.ncei.noaa.gov/data/local-climatological-data/

Table 6: **Standard deviations of the metrics in Table. 2 and Table. 3**.

| Method / H | MSE | | | | | | | | | | | |
|---|---|---|---|---|---|---|---|---|---|---|---|---|
| | ETTH2 | | | ETTm1 | | | WTH | | | ECL | | |
| | 1 | 24 | 48 | 1 | 24 | 48 | 1 | 24 | 48 | 1 | 24 | 48 |
| Informer | 1.370 | 2.254 | 2.088 | 0.088 | 0.035 | 0.020 | 0.005 | 0.003 | 0.009 | | | |
| OnlineTCN | 0.011 | 0.017 | 0.148 | 0.003 | 0.002 | 0.002 | 0.001 | 0.001 | 0.001 | 0.019 | 0.077 | 0.122 |
| TFCL | 0.030 | 0.005 | 0.279 | 0.004 | 0.006 | 0.010 | 0.002 | 0.001 | 0.004 | 0.047 | 0.338 | 0.253 |
| ER | 0.018 | 0.007 | 0.141 | 0.005 | 0.003 | 0.004 | 0.001 | 0.001 | 0.009 | 0.034 | 0.236 | 0.320 |
| MIR | 0.019 | 0.017 | 0.130 | 0.005 | 0.005 | 0.006 | 0.002 | 0.001 | 0.009 | 0.037 | 0.261 | 0.143 |
| DER++ | 0.022 | 0.024 | 0.143 | 0.003 | 0.002 | 0.003 | 0.001 | 0.001 | 0.011 | 0.027 | 0.072 | 0.146 |
| FSNet | 0.018 | 0.014 | 0.128 | 0.003 | 0.002 | 0.003 | 0.001 | 0.001 | 0.001 | 0.021 | 0.096 | 0.105 |
| Time-TCN | 0.020 | 0.010 | 0.189 | 0.004 | 0.004 | 0.005 | 0.001 | 0.001 | 0.001 | 0.033 | 0.130 | 0.232 |
| PatchTST | 0.022 | 0.010 | 0.183 | 0.005 | 0.005 | 0.007 | 0.002 | 0.001 | 0.007 | 0.039 | 0.167 | 0.239 |
| OneNet-TCN | 0.015 | 0.012 | 0.104 | 0.003 | 0.003 | 0.003 | 0.001 | 0.001 | 0.007 | 0.025 | 0.114 | 0.152 |
| OneNet | 0.015 | 0.014 | 0.100 | 0.003 | 0.002 | 0.003 | 0.001 | 0.001 | 0.005 | 0.021 | 0.086 | 0.099 |

| Method / H | MAE | | | | | | | | | | | |
|---|---|---|---|---|---|---|---|---|---|---|---|---|
| | ETTH2 | | | ETTm1 | | | WTH | | | ECL | | |
| | 1 | 24 | 48 | 1 | 24 | 48 | 1 | 24 | 48 | 1 | 24 | 48 |
| Informer | 0.043 | 0.102 | 0.091 | 0.060 | 0.023 | 0.014 | 0.005 | 0.003 | 0.008 | | | |
| OnlineTCN | 0.007 | 0.002 | 0.016 | 0.002 | 0.002 | 0.003 | 0.002 | 0.077 | 0.122 | 0.002 | 0.009 | 0.011 |
| TFCL | 0.003 | 0.003 | 0.024 | 0.008 | 0.005 | 0.008 | 0.002 | 0.001 | 0.006 | 0.011 | 0.019 | 0.008 |
| ER | 0.017 | 0.006 | 0.013 | 0.009 | 0.002 | 0.004 | 0.002 | 0.001 | 0.005 | 0.011 | 0.017 | 0.014 |
| MIR | 0.018 | 0.005 | 0.012 | 0.009 | 0.004 | 0.005 | 0.002 | 0.001 | 0.005 | 0.013 | 0.013 | 0.012 |
| DER++ | 0.015 | 0.004 | 0.015 | 0.007 | 0.002 | 0.002 | 0.002 | 0.001 | 0.007 | 0.002 | 0.013 | 0.014 |
| FSNet | 0.009 | 0.005 | 0.012 | 0.004 | 0.002 | 0.002 | 0.001 | 0.001 | 0.001 | 0.001 | 0.011 | 0.011 |
| Time-TCN | 0.009 | 0.004 | 0.018 | 0.006 | 0.003 | 0.005 | 0.002 | 0.026 | 0.044 | 0.008 | 0.009 | 0.011 |
| PatchTST | 0.013 | 0.005 | 0.016 | 0.009 | 0.004 | 0.006 | 0.002 | 0.001 | 0.005 | 0.012 | 0.010 | 0.011 |
| OneNet-TCN | 0.017 | 0.005 | 0.013 | 0.008 | 0.003 | 0.004 | 0.002 | 0.001 | 0.006 | 0.009 | 0.009 | 0.013 |
| OneNet | 0.014 | 0.005 | 0.013 | 0.007 | 0.003 | 0.003 | 0.002 | 0.001 | 0.004 | 0.005 | 0.007 | 0.012 |

- Learning rate $3e-3$ on Traffic and ECL and $1e-3$ for other datasets. Learning rate $1e-2$ for the EGD algorithm and $1e-3$ for the offline reinforcement learning block, where the selection scope is $\{1e-3, 3e-3, 1e-2, 3e-2\}$.

- Number of hidden layers 10 for both cross-time and cross-variable branches, where the selection scope is $\{6, 8, 10, 12\}$.

- Adapter's EMA coefficient 0.9, Gradient EMA for triggering the memory interaction 0.3, where the selection scope is $\{0.1, 0.2, \ldots, 1.0\}$.

- Memory triggering threshold 0.75, where the selection scope is $\{0.6, 0.65, 0.7 \ldots, 0.9\}$.

- Episodic memory size: 5000 (for ER, MIR, and DER++), 50 (for TFCL).

### C.5  Additional Numerical Results

**Additional forecasting results.** In this section, we analyze the performance of different forecasting methods on four datasets, ECL, WTH, ETTh2, and ETTm1, with various starting points, as shown in Figure 8, Figure 9, Figure 10, and Figure 11, respectively. For the last three datasets, all methods produce similar results that can capture the underlying time series patterns, and the performance differences are not significant. However, when it comes to the ECL dataset, we observe that almost all baselines exhibit poor forecasting results at the onset of the concept shift (time step 2500). As we provide more instances, the performance of these methods improves, as evidenced by the cumulative loss curves in Figure 7 and Figure 6.

**Abltion studies of hyper-parameters.** We conduct detailed ablation studies about model layers, learning rate, and model dimension here. Taking into account the learning rate for the two-branch framework, the learning rate for the long-term weight, and the learning rate for the short-term weight: $lr, lr_{\mathbf{w}}, lr_{\mathbf{b}}$, as shown in Table 10 (left), the impact of the learning rate on dual-stream

Table 7: **MSE and MAE of various adaptation methods**. H: forecast horizon. OneNet-TCN+Patch is the mixture of TCN, Time-TCN, and PatchTST.

| Metric | Method | ETTH2 | | | ETTm1 | | | WTH | | | ECL | | | Avg |
|---|---|---|---|---|---|---|---|---|---|---|---|---|---|---|
| | | 1 | 24 | 48 | 1 | 24 | 48 | 1 | 24 | 48 | 1 | 24 | 48 | |
| MSE | TCN | 0.502 | 0.830 | 1.183 | 0.214 | 0.258 | 0.283 | 0.206 | 0.308 | 0.302 | 3.309 | 11.339 | 11.534 | 2.522 |
| | Time-TCN | 0.491 | 0.779 | 1.307 | 0.093 | 0.281 | 0.308 | 0.158 | 0.311 | 0.308 | 4.060 | 5.260 | 5.230 | 1.549 |
| | PatchTST | 0.362 | 1.622 | 2.716 | 0.083 | 0.427 | 0.553 | 0.162 | 0.372 | 0.465 | 2.022 | 4.325 | 5.030 | 1.512 |
| | OneNet-TCN | 0.411 | 0.772 | 0.806 | 0.082 | 0.212 | 0.223 | 0.171 | 0.293 | 0.310 | 2.470 | 4.713 | 4.567 | 1.253 |
| | OneNet-TCN+Patch | 0.355 | 0.844 | 1.120 | 0.079 | 0.239 | 0.255 | 0.163 | 0.298 | 0.314 | 2.172 | 4.142 | 4.149 | 1.178 |
| MAE | TCN | 0.436 | 0.547 | 0.589 | 0.085 | 0.381 | 0.403 | 0.276 | 0.367 | 0.362 | 0.635 | 1.196 | 1.235 | 0.543 |
| | Time-TCN | 0.425 | 0.544 | 0.636 | 0.211 | 0.395 | 0.421 | 0.204 | 0.378 | 0.378 | 0.332 | 0.420 | 0.438 | 0.399 |
| | PatchTST | 0.341 | 0.577 | 0.672 | 0.186 | 0.471 | 0.549 | 0.200 | 0.393 | 0.459 | 0.224 | 0.341 | 0.375 | 0.399 |
| | OneNet-TCN | 0.374 | 0.511 | 0.543 | 0.191 | 0.319 | 0.371 | 0.221 | 0.345 | 0.356 | 0.411 | 0.513 | 0.534 | 0.391 |
| | OneNet-TCN+Patch | 0.338 | 0.513 | 0.552 | 0.184 | 0.360 | 0.381 | 0.217 | 0.351 | 0.381 | 0.297 | 0.423 | 0.457 | 0.371 |

networks is quite significant. The optimal learning rate varies for each dataset, but we can see that for each dataset, the optimal learning rate is generally within the range of $[1e − 4, 1e − 2]$. $lr_{\mathbf{w}}$ has a relatively small impact on the final performance of the model. On the contrary, the offline-RL module determines whether the weights can quickly adapt to the new distribution, which has a greater impact on the final performance. In terms of model parameters, # Layers, $d_m$, and $d_{head}$, all three have a significant impact on the performance of the model. A small model may not be able to fit the training data, but a model that is too large increases the risk of overfitting, so each dataset has an optimal model size. However, in this paper, we use the same hyperparameters for all datasets to simplify the complexity of training and model selection. Specifically, we set $lr = 1e − 3, lr_{\mathbf{w}} = 1e − 2, lr_b = 1e − 3, \#layers = 10, d_m = 64, d_{head} = 320$.

### C.6 Ensembling more than two networks.

In the main paper, we verify the effectiveness of ensembling two branches with different model biases. Here, we show that the proposed OneNet framework enables us to incorporate more branches and the OCP block can fully utilize the benefit of each branch. As shown in Table 7, incorporating PatchTST to OneNet-TCN will further reduce the forecasting results during online forecasting.

### C.7 More visualization results and Convergence Analysis of Different Structures

As shown in Figure 6 and Figure 7, ETTh2 and ECL datasets pose the greatest challenge to all models due to the sharp peaks in their loss curves. When the forecasting window is short, OneNet outperforms all baselines by a significant margin on all datasets. When the forecasting window is extended to $H = 48$, FSNet is comparable to OneNet in the first three datasets. However, when concept drift occurs in the ECL dataset, all baselines experience a drastic increase in their cumulative MSE, except OneNet, which maintains a low MSE. Furthermore, the initialized MSE error of OneNet is consistently lower than that of all baselines, thanks to the two-stream structure of OneNet. For instance, in Figure 6(b) and Figure 6(f), OneNet demonstrates a significantly lower MSE than baselines when the number of instances is less than 100.

### C.8 Online forecasting results with delayed feedback

As illustrated in Section 2, this paper adopts the same setting as FSNet [Pham et al., 2023], where the true values of each time step are revealed to improve the performance of the model in subsequent rounds. However, in real-world applications, the true values of the forecast horizon $H$ may not be available until $H$ rounds later, which is known as online forecasting with delayed feedback. This setting is more challenging because the model cannot be retrained at each round and we can only train the model per $H$ round. Tables 8 and 9 show the cumulative performance considering MSE and MAE, respectively. As expected, all methods perform worse with delayed feedback than under the traditional online forecasting setting. Notably, the state-of-the-art method FSNet is shown to be sensitive to delayed feedback, particularly when $H = 48$, where it is even inferior to a simple TCN baseline on some datasets. In contrast, our proposed method OneNet significantly outperforms all continual learning baselines across different datasets and delayed forecast horizons.

Table 8: **MSE of various adaptation methods with delayed feedback**. H: forecast horizon. OneNet-TCN is the mixture of TCN and Time-TCN, and OneNet is the mixture of FSNet and Time-FSNet.

| Method / H | ETTH2 | | | ETTm1 | | | WTH | | | ECL | | | |
|---|---|---|---|---|---|---|---|---|---|---|---|---|---|
| | 1 | 24 | 48 | 1 | 24 | 48 | 1 | 24 | 48 | 1 | 24 | 48 | Avg |
| OnlineTCN | 0.502 | 5.871 | 11.074 | 0.214 | 0.410 | 0.535 | 0.206 | 0.429 | 0.504 | 3.309 | 9.621 | 24.159 | 4.736 |
| ER | 0.508 | 5.461 | 17.329 | 0.086 | 0.367 | 0.498 | 0.180 | 0.373 | 0.435 | 2.579 | 8.091 | 17.700 | 4.467 |
| DER++ | 0.508 | 5.387 | 17.334 | 0.083 | 0.347 | 0.465 | 0.174 | 0.369 | 0.431 | 2.657 | 7.878 | 17.692 | 4.444 |
| FSNet | 0.466 | 5.765 | 11.907 | 0.085 | 0.383 | 0.502 | 0.162 | 0.335 | 0.411 | 3.143 | 8.722 | 27.150 | 4.919 |
| OneNet-TCN | 0.411 | 2.639 | 4.995 | 0.082 | **0.287** | 0.382 | 0.171 | 0.341 | 0.433 | 2.470 | **4.809** | 6.252 | 1.939 |
| OneNet | **0.380** | **2.064** | **4.952** | **0.082** | 0.332 | **0.351** | **0.156** | **0.323** | **0.394** | **2.351** | 4.984 | **6.226** | **1.883** |

Table 9: **MAE of various adaptation methods with delayed feedback**. H: forecast horizon. OneNet-TCN is the mixture of TCN and Time-TCN, and OneNet is the mixture of FSNet and Time-FSNet.

| Method / H | ETTH2 | | | ETTm1 | | | WTH | | | ECL | | | |
|---|---|---|---|---|---|---|---|---|---|---|---|---|---|
| | 1 | 24 | 48 | 1 | 24 | 48 | 1 | 24 | 48 | 1 | 24 | 48 | Avg |
| OnlineTCN | 0.436 | 1.109 | 1.348 | 0.085 | 0.511 | 0.548 | 0.276 | 0.459 | 0.508 | 0.635 | 0.783 | 1.076 | 0.648 |
| ER | 0.376 | 0.976 | 1.651 | 0.197 | 0.456 | 0.525 | 0.244 | 0.421 | 0.459 | 0.506 | 0.595 | 0.772 | 0.598 |
| DER++ | 0.375 | 0.967 | 1.644 | 0.192 | 0.443 | 0.508 | 0.235 | 0.415 | 0.456 | 0.421 | 0.591 | 0.758 | 0.584 |
| FSNet | 0.368 | 0.983 | 1.494 | 0.191 | 0.468 | 0.502 | 0.216 | 0.394 | 0.453 | 0.472 | 0.827 | 1.391 | 0.554 |
| OneNet-TCN | 0.374 | 0.772 | 0.951 | 0.191 | **0.387** | **0.417** | 0.221 | 0.389 | 0.461 | 0.411 | **0.381** | 0.451 | 0.451 |
| OneNet | **0.348** | **0.684** | **0.916** | **0.187** | 0.428 | 0.430 | **0.201** | **0.381** | **0.436** | **0.254** | 0.387 | **0.444** | **0.425** |

Table 10: **Results of different OneNet 's hyper-parameter configurations** on the benchmarks ($H = 48$). $lr, lr_{\mathbf{w}}, lr_{\mathbf{b}}$ are the learning rate for the two-branch framework, the learning rate for the long-term weight, and the learning rate for the short-term weight. # Layers is the number of layers of the two branches of OneNet. $d_m, d_{head}$ is the hidden dimension and the output dimension of the encoders, respectively.

| Hyper-Parameter | Value | MSE | | | | Hyper-Parameter | Value | MSE | | | |
|---|---|---|---|---|---|---|---|---|---|---|---|
| | | ETTh2 | ETTm1 | WTH | ECL | | | ETTh2 | ETTm1 | WTH | ECL |
| $lr$ | 1.00E-01 | - | - | - | - | # Layers | 6 | 0.632 | 0.114 | 0.203 | 2.402 |
| | 1.00E-02 | 0.585 | 0.152 | 0.171 | 3.128 | | 8 | 0.661 | 0.101 | 0.201 | 2.289 |
| | 1.00E-03 | 0.656 | 0.111 | 0.196 | 2.516 | | 10 | 0.609 | 0.108 | 0.200 | 2.201 |
| | 1.00E-04 | 2.994 | 0.464 | 0.331 | 4.949 | | 12 | 0.652 | 0.115 | 0.200 | 2.328 |
| $lr_{\mathbf{w}}$ | 1.00E-01 | 0.619 | 0.108 | 0.202 | 2.177 | $d_m$ | 16 | 0.679 | 0.122 | 0.223 | 2.201 |
| | 1.00E-02 | 0.609 | 0.108 | 0.205 | 2.184 | | 32 | 0.612 | 0.116 | 0.210 | 2.810 |
| | 1.00E-03 | 0.608 | 0.108 | 0.201 | 2.197 | | 64 | 0.609 | 0.108 | 0.200 | 2.311 |
| | 1.00E-04 | 0.607 | 0.108 | 0.201 | 2.197 | | 160 | 0.619 | 0.108 | 0.200 | 2.141 |
| $lr_{\mathbf{b}}$ | 1.00E-01 | 0.899 | 0.134 | 0.221 | 2.499 | $d_{head}$ | 80 | 0.741 | 0.136 | 0.219 | 2.468 |
| | 1.00E-02 | 0.876 | 0.112 | 0.197 | 2.372 | | 160 | 0.600 | 0.112 | 0.214 | 2.364 |
| | 1.00E-03 | 0.656 | 0.111 | 0.196 | 2.371 | | 320 | 0.609 | 0.108 | 0.201 | 2.184 |
| | 1.00E-04 | 0.643 | 0.111 | 0.196 | 2.362 | | 500 | 0.571 | 0.104 | 0.182 | 2.182 |

Table 11: **Ablation studies of the variable independence and frequency domain augmentation**, where the metric is MSE. FEDformer-F uses frequency-enhanced blocks with Fourier transform, and FEDformer-W uses frequency-enhanced blocks with Wavelet transform. Time-TCN is the variable independence version of TCN.

| Method | Online | ETTH2 | | | ETTm1 | | | WTH | | | ECL | | | Avg |
|---|---|---|---|---|---|---|---|---|---|---|---|---|---|---|
| | | 1 | 24 | 48 | 1 | 24 | 48 | 1 | 24 | 48 | 1 | 24 | 48 | |
| FEDformer-F | ✗ | 1.922 | 3.045 | 4.016 | 0.922 | 1.003 | 1.821 | 3.544 | 2.344 | 1.179 | 43.852 | 37.802 | 37.377 | 11.569 |
| | ✓ | 1.912 | 3.013 | 3.951 | 0.372 | 0.633 | 0.586 | 2.196 | 0.376 | 0.562 | 39.243 | 35.975 | 36.092 | 10.409 |
| FEDformer-W | ✗ | 1.816 | 3.070 | 3.996 | 2.275 | 3.784 | 2.662 | 1.220 | 1.211 | 1.431 | 41.791 | 37.236 | 37.210 | 11.475 |
| | ✓ | 1.798 | 2.993 | 1.623 | 0.235 | 0.451 | 0.516 | 0.717 | 0.962 | 0.372 | 21.387 | 24.600 | 27.640 | 6.941 |
| TCN | ✗ | 27.060 | 27.760 | 26.320 | 2.240 | 12.170 | 10.880 | 0.290 | 0.480 | 0.580 | 538.000 | 546.000 | 552.000 | 145.315 |
| | ✓ | 0.530 | 0.930 | 0.910 | 0.130 | 0.310 | 0.250 | 0.300 | 0.348 | 0.348 | 3.010 | 11.680 | 10.800 | 2.462 |
| Time-TCN | ✗ | 4.530 | 7.840 | 11.017 | 0.097 | 0.800 | 11.017 | 0.162 | 0.344 | 0.429 | 47.900 | 48.660 | 67.150 | 15.020 |
| | ✓ | 0.480 | 0.780 | 0.867 | 0.090 | 0.280 | 0.310 | 0.300 | 0.310 | 0.309 | 4.010 | 5.220 | 5.210 | 1.550 |

Table 12: **Comparison of existing forecasting structures**, including TCN [Bai et al., 2018], FED-former [Zhou et al., 2022b], PatchTST [Nie et al., 2023], Dlinear [Zeng et al., 2023], Nlinear [Zeng et al., 2023], TS-Mixer [Chen et al., 2023], and CrossFormer [Zhang and Yan, 2023].

| Method | Online | ETTH2 | | | ETTm1 | | | WTH | | | ECL | | | Avg |
|---|---|---|---|---|---|---|---|---|---|---|---|---|---|---|
| | | 1 | 24 | 48 | 1 | 24 | 48 | 1 | 24 | 48 | 1 | 24 | 48 | |
| FEDformer-F | ✗ | 1.922 | 3.045 | 4.016 | 0.922 | 1.003 | 1.821 | 3.544 | 2.344 | 1.179 | 43.852 | 37.802 | 37.377 | 11.569 |
| | ✓ | 1.912 | 3.013 | 3.951 | 0.372 | 0.633 | 0.586 | 2.196 | 0.376 | 0.562 | 39.243 | 35.975 | 36.092 | 10.409 |
| FEDformer-W | ✗ | 1.816 | 3.070 | 3.996 | 2.275 | 3.784 | 2.662 | 1.220 | 1.211 | 1.431 | 41.791 | 37.236 | 37.210 | 11.475 |
| | ✓ | 1.798 | 2.993 | 1.623 | 0.235 | 0.451 | 0.516 | 0.717 | 0.962 | 0.372 | 21.387 | 24.600 | 27.640 | 6.941 |
| PatchTST | ✗ | 0.427 | 2.090 | 3.290 | 0.083 | 0.433 | 0.570 | 0.163 | 0.375 | 0.467 | 2.030 | 4.395 | 5.101 | 1.619 |
| | ✓ | 0.362 | 1.622 | 2.716 | 0.083 | 0.427 | 0.553 | 0.162 | 0.372 | 0.465 | 2.022 | 4.325 | 5.030 | 1.512 |
| Crossformer | ✗ | 23.270 | 28.904 | 29.218 | 0.400 | 1.433 | 1.691 | 0.146 | 0.327 | 0.426 | 469.260 | 475.490 | 478.270 | 125.736 |
| | ✓ | 9.873 | 2.856 | 5.772 | 0.096 | 0.356 | 0.370 | 0.149 | 0.317 | 0.359 | 68.300 | 92.500 | 94.790 | 22.978 |
| TCN | ✗ | 27.060 | 27.760 | 26.320 | 2.240 | 12.170 | 10.880 | 0.290 | 0.480 | 0.580 | 538.000 | 546.000 | 552.000 | 145.315 |
| | ✓ | 0.530 | 0.930 | 0.910 | 0.130 | 0.310 | 0.250 | 0.300 | 0.348 | 0.348 | 3.010 | 11.680 | 10.800 | 2.462 |
| Time-TCN | ✗ | 4.530 | 7.840 | 1.300 | 0.097 | 0.800 | 1.030 | 0.162 | 0.344 | 0.429 | 47.900 | 48.660 | 67.150 | 15.020 |
| | ✓ | 0.480 | 0.780 | 1.300 | 0.090 | 0.280 | 0.310 | 0.300 | 0.310 | 0.309 | 4.010 | 5.220 | 5.210 | 1.550 |
| DLinear | ✗ | 2.91 | 10.25 | 7.53 | 0.538 | 1.461 | 1.233 | 0.266 | 0.462 | 0.542 | 12.03 | 51.28 | 58.46 | 12.247 |
| | ✓ | 2.44 | 9.24 | 6.91 | 0.46 | 1.3 | 1.12 | 0.262 | 0.459 | 0.541 | 6.69 | 27.82 | 31.54 | 7.399 |
| NLinear | ✗ | 0.424 | 50.15 | 49.52 | 0.09 | 4.02 | 4.13 | 0.171 | 1.07 | 1.08 | 2.14 | 930 | 929 | 164.316 |
| | ✓ | 0.369 | 50.24 | 49.6 | 0.089 | 4.035 | 4.141 | 0.171 | 1.053 | 1.064 | 2.135 | 930 | 930 | 164.408 |
| TS-Mixer | ✗ | 1.968 | 3.525 | 4.88 | 0.335 | 0.726 | 0.855 | 0.255 | 0.429 | 0.503 | 11.16 | 30.93 | 44.68 | 8.354 |
| | ✓ | 0.78 | 2.05 | 3.060 | 0.219 | 0.550 | 0.660 | 0.237 | 0.413 | 0.482 | 2.798 | 4.983 | 5.764 | 1.833 |

## C.9 The effect of variable independence and frequency domain augmentation

As shown in Table 11, we observe that frequency-enhanced blocks, which use the wavelet transform, offer greater robustness to the Fourier transform. FEDformer outperforms TCN in terms of generalization, but online adaptation has a limited impact on performance, similar to other transformer-based models. Notably, we find that variable independence is crucial for model robustness. By convolving solely on the time dimension, independent of the feature channel, we significantly reduce MSE error compared to convolving on the feature channel, regardless of whether online adaptation is applied.

## C.10 Comparison of existing forecasting structures.

Results are shown in Table 12. Considering the average MSE on all four datasets, all transformer-based models and Dlinear are better than TCN and Time-TCN. However, with online adaptation, the forecasting error of TCN structures is reduced by a large margin and is better than DLinear and FEDformer. Specifically, we show that the current transformer-based model (PatchTST [Nie et al., 2023]) demonstrates superior generalization performance than the TCN models **even without any online adaptation**, particularly in the challenging ECL task. However, we also noticed that PatchTST remains largely unchanged after online retraining. In contrast, the TCN structure can quickly adapt to the shifted distribution, and the online updated TCN model prefers a better forecasting error than the adapted PatchTST on the first three data sets. Therefore, it is promising to combine the strengths of both structures to create a more robust and adaptable model that can handle shifting data distributions better.

**Algorithm 1** Training and inference algorithm of OneNet

1: **Input:** Historical multivariate time series $\mathbf{x} \in \mathbb{R}^{M \times L}$ with $M$ variables and length $L$, the forecast target $\mathbf{y} \in \mathbb{R}^{M \times H}$, where we omit the variable index and time step index for simplicity.

2: **Initialize** a cross-time forecaster $f_1$, a cross-variable forecaster $f_2$ with corresponding prediction head, long-term weight $\mathbf{w} = [0.5, 0.5]$, short term learning block $f_{rl} : \mathbb{R}^{H \times M \times 3} \to \mathbb{R}^2$, and step size $\eta$ for long-term weight updating.

3: **Get prediction results from two forecasters.**

4: $\tilde{\mathbf{y}}_1 \in \mathbb{R}^{M \times H} = f_1(\mathbf{x})$.        *// Prediction result from the cross-time forecaster.*

5: $\tilde{\mathbf{y}}_2 \in \mathbb{R}^{M \times H} = f_2(\mathbf{x})$.        *// Prediction result from the cross-variable forecaster.*

6: **Get combination weight from the OCP block.**

7: $\mathbf{b} = f_{rl}\left(w_1 \tilde{\mathbf{y}}_1 \otimes w_2 \tilde{\mathbf{y}}_2 \otimes \mathbf{y}\right)$      *// Calculate the short term weight.*

8: $\tilde{w}_i = (w_i + b_i)/\left(\sum_{i=1}^{d}(w_i + b_i)\right)$    *// Calculate the normalized weight.*

9: $\tilde{\mathbf{y}} = w_1 * \tilde{\mathbf{y}}_1 + w_2 * \tilde{\mathbf{y}}_2$.        *// The final prediction result.*

10: **Update the long/short term weight.**

11: $w_i = w_i \exp(-\eta \parallel \tilde{\mathbf{y}}_i - \mathbf{y} \parallel^2)/\left(\sum_{i=1}^{2} w_i \exp(-\eta \parallel \tilde{\mathbf{y}}_i - \mathbf{y} \parallel^2)\right)$

12: $f_{rl} \leftarrow \text{Adam}\left(f_{rl}, \mathcal{L}(w_1 * \tilde{\mathbf{y}}_1 + w_2 * \tilde{\mathbf{y}}_2, \mathbf{y})\right)$     *//Parameters such as learning rate are omitted.*

13: **Update the two forecasters.**

14: $f_1 \leftarrow \text{Adam}\left(f_1, \mathcal{L}(\tilde{\mathbf{y}}_1, \mathbf{y})\right)$, $f_2 \leftarrow \text{Adam}\left(f_2, \mathcal{L}(\tilde{\mathbf{y}}_2, \mathbf{y})\right)$

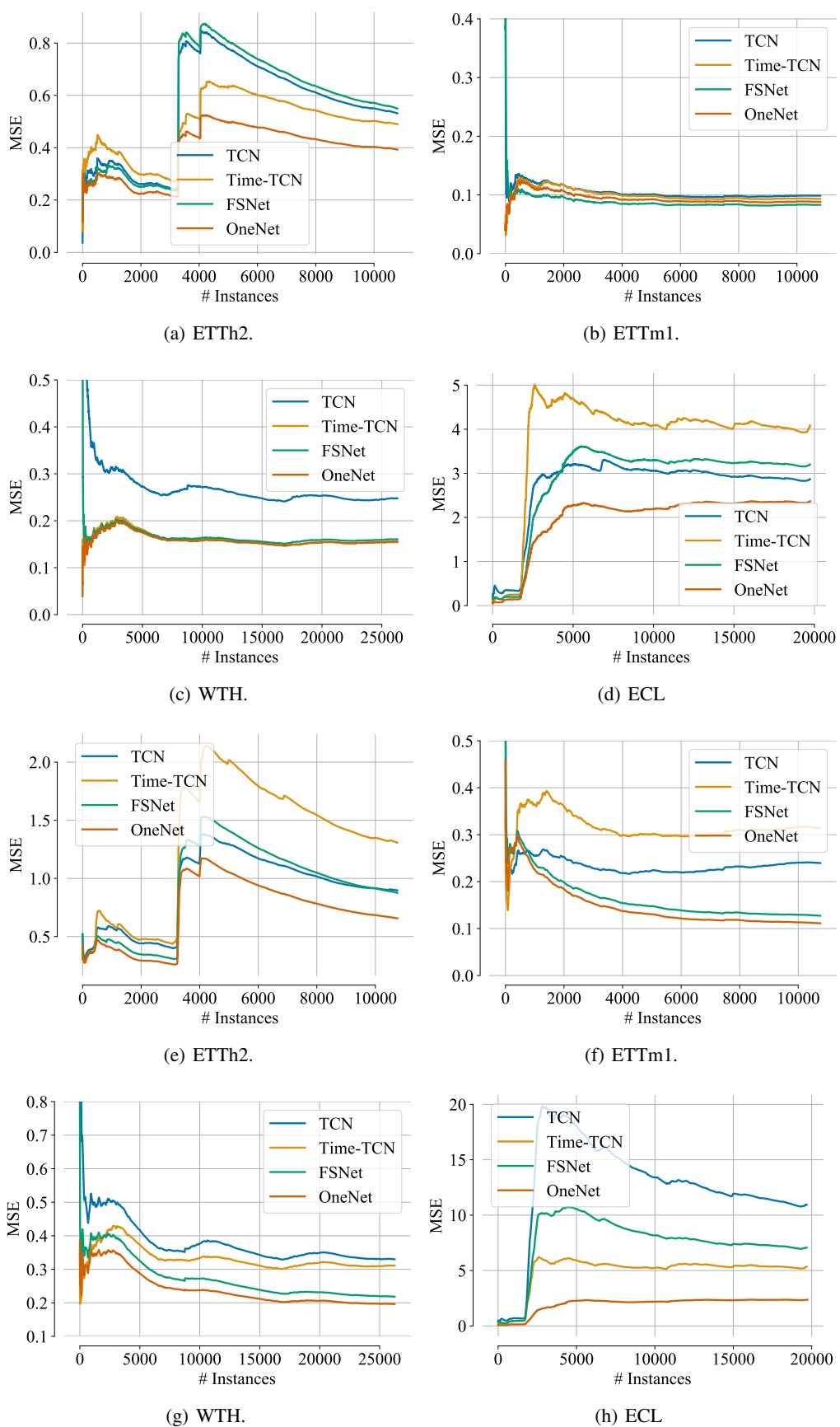

Figure 6: **Evolution of the cumulative MSE loss** during training with forecast window $H = 1$ (a, b, c, d) and $H = 48$ (e, f, g, h).

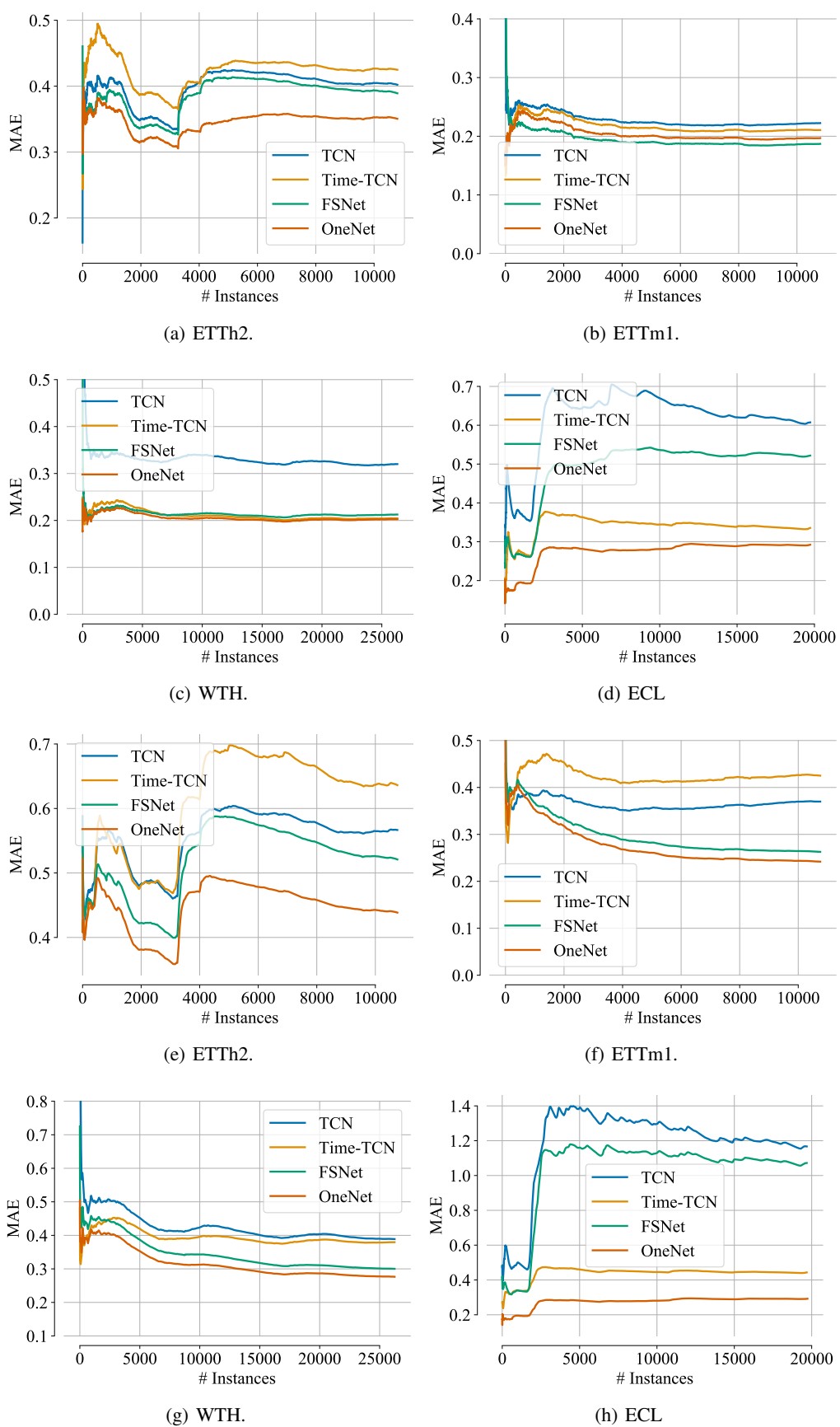

Figure 7: Evolution of the cumulative MAE loss during training with forecasting window $H = 1$ (a,b,c,d) and $H = 48$ (e,f,g,h).

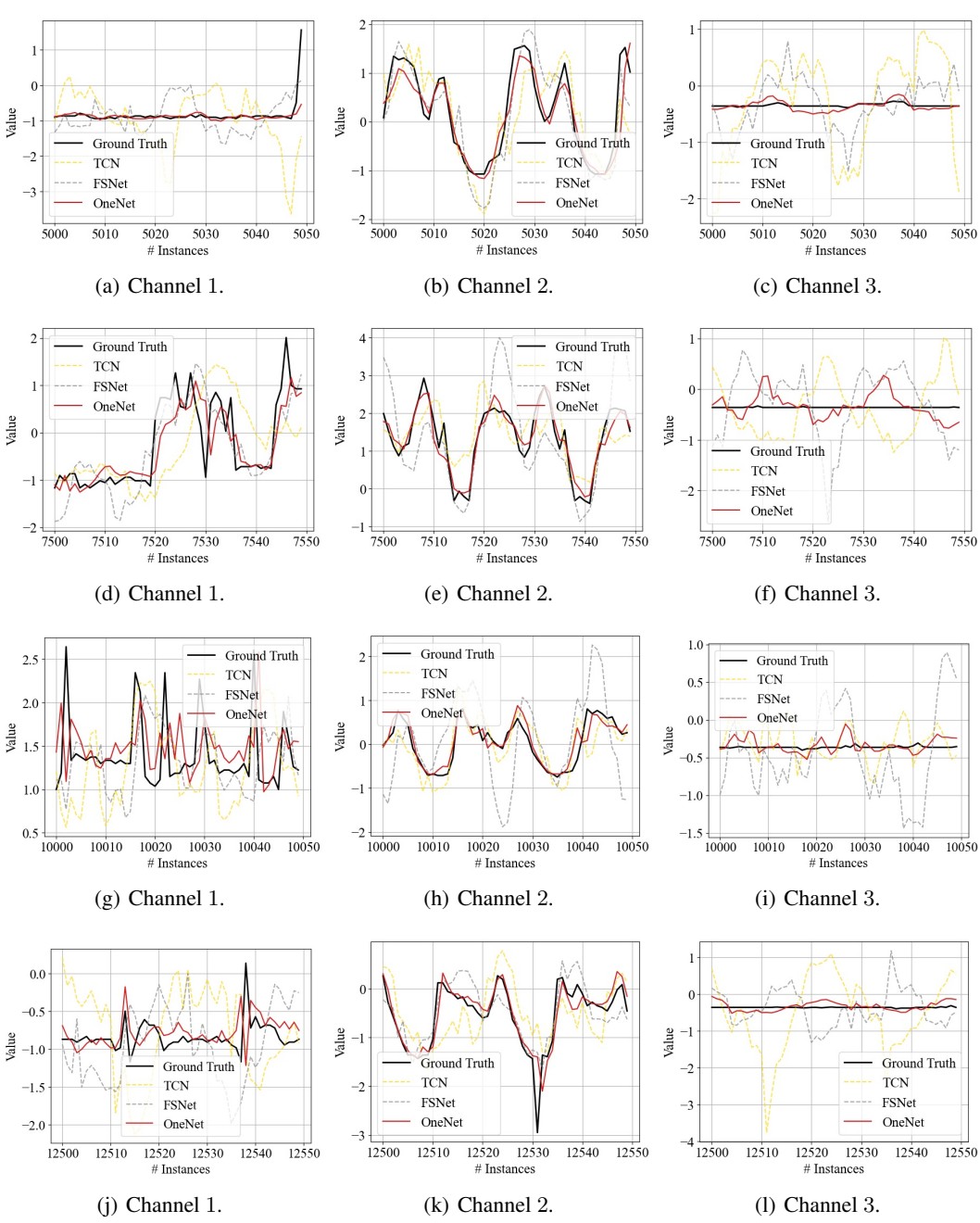

Figure 8: **Visualization of the model's prediction throughout the online learning process** in the ECL dataset. We focus on a short horizon of 50 time steps and the start prediction time is from 5000 (a,b,c), 7500 (d,e,f), 10000 (g,h,i), and 12500 (j,k,l) respectively.

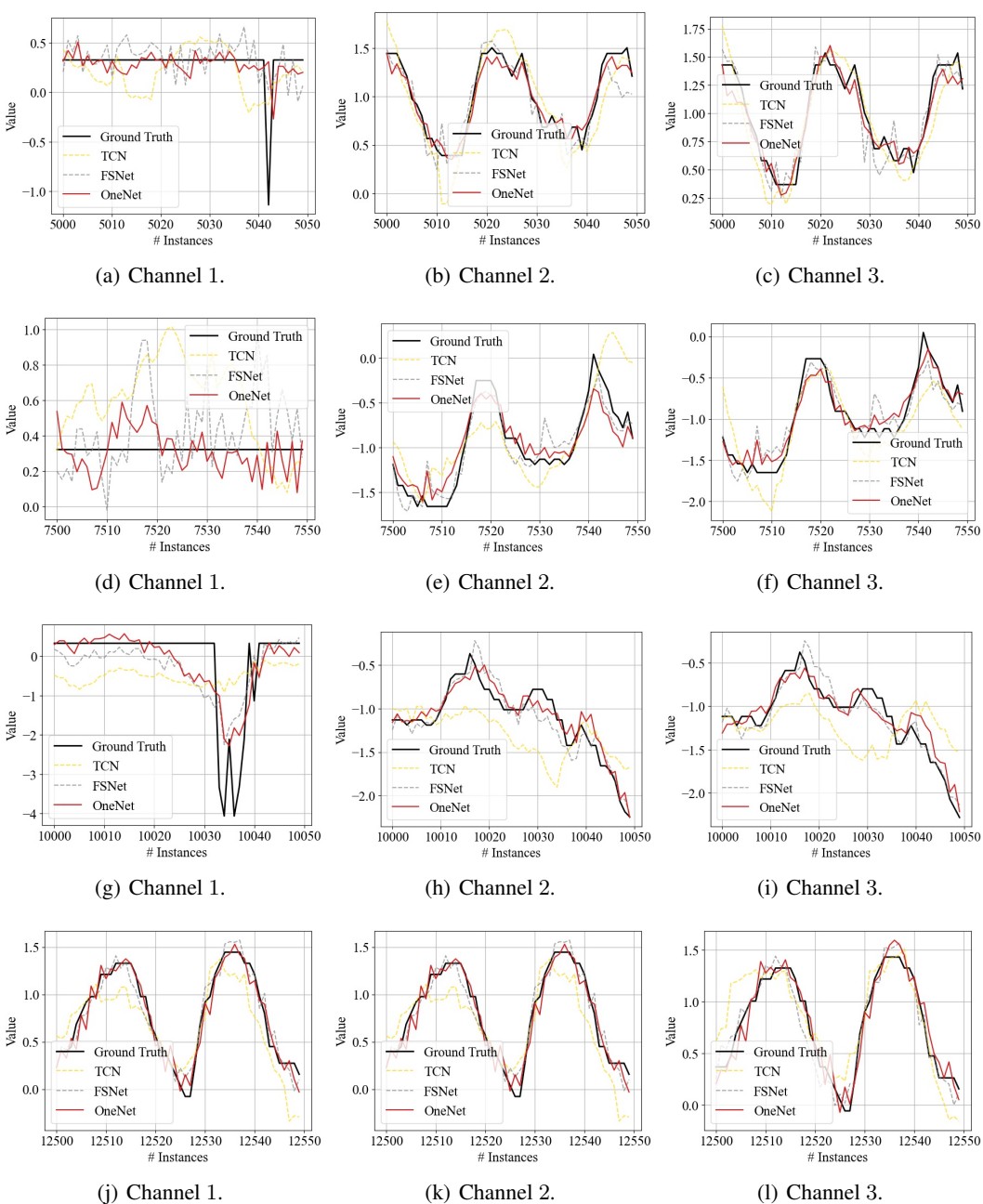

Figure 9: **Visualization of the model's prediction throughout the online learning process** on the WTH dataset. We focus on a short horizon of 50 time steps and the start prediction time is from 5000 (a,b,c), 7500 (d,e,f), 10000 (g,h,I), and 12500 (j,k,l), respectively.

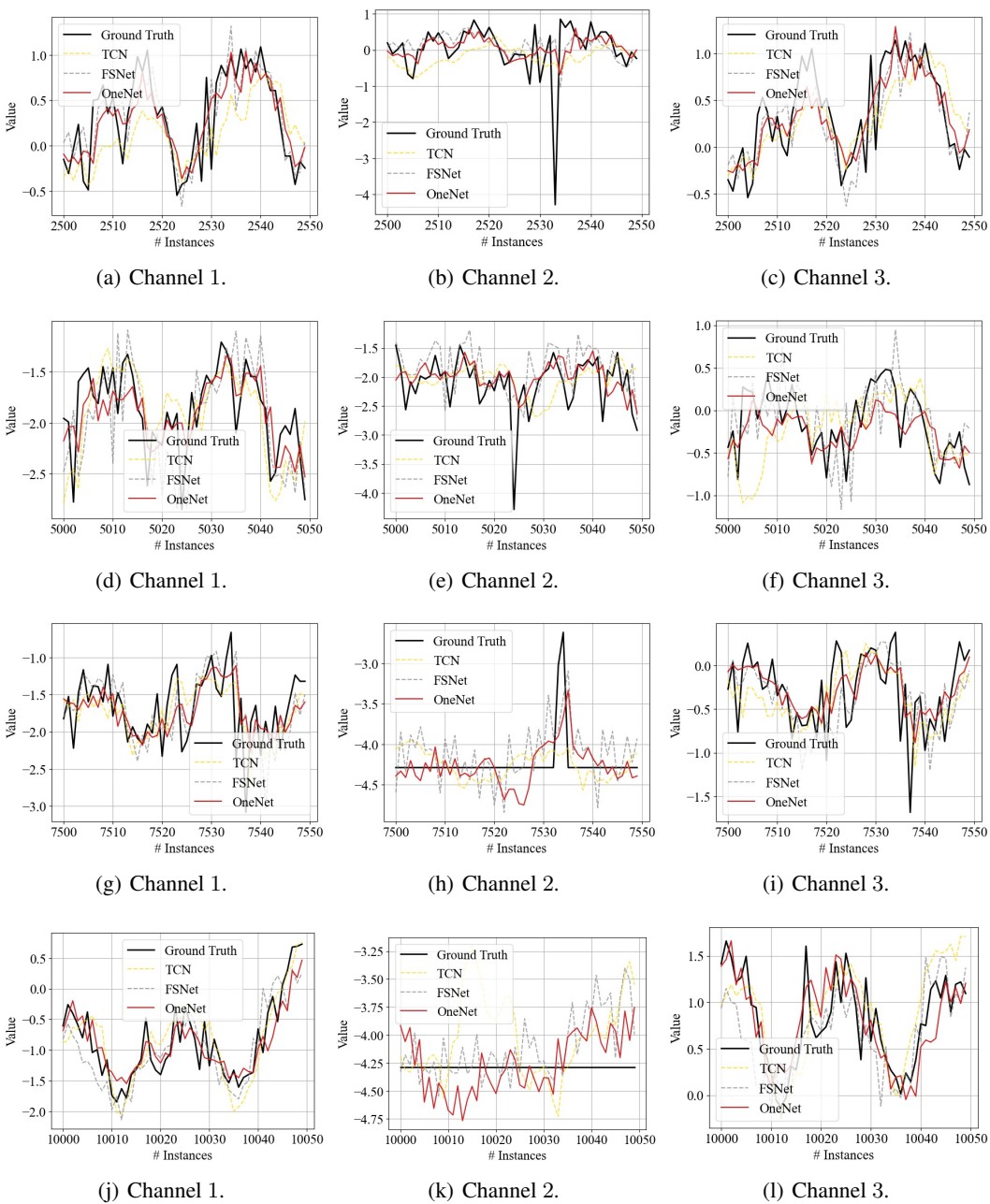

Figure 10: **Visualization of the model's prediction throughout the online learning process** in the ETTh2 data set. We focus on a short horizon of 50 time steps and the start prediction time is from 2500 (a,b,c), 5000 (d,e,f), 7500 (g,h,i), and 10000 (j,k,l) respectively.

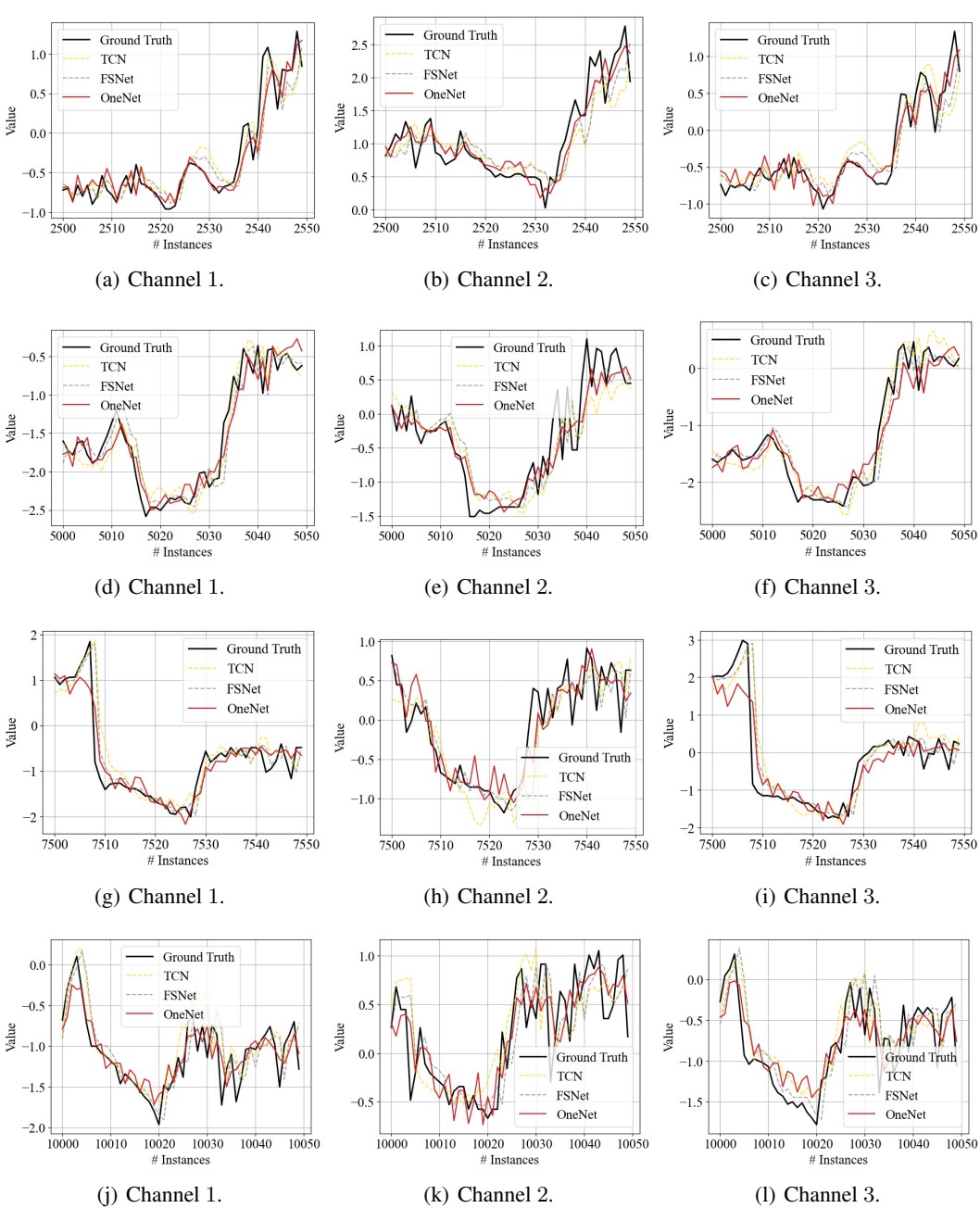

Figure 11: **Visualization of the model's prediction throughout the online learning process** on the ETTm1 dataset. We focus on a short horizon of 50 time steps and the start prediction time is from 2500 (a, b, c), 5000 (d, e, f), 7500 (g, h, i) and 10000 (j, k, l), respectively.

