# OpenReview forum: "OneNet: Enhancing Time Series Forecasting Models under Concept Drift by Online Ensembling"
_NeurIPS.cc/2023/Conference — NeurIPS 2023 poster_

### Official Review · Reviewer_UB5d · 2023-06-25

**Soundness:** 2 fair
**Presentation:** 2 fair
**Contribution:** 2 fair
**Rating:** 5
**Confidence:** 4

**Summary:**

The paper proposes a method for online learning based on ensemble learning. The method proposed by the authors leverages cross-variable and cross-time dependency. In addition, the authors propose a reinformed learning method that dynamically adjusts the weights used to combine the models.

**Strengths:**

See Question section

**Weaknesses:**

See Question section

**Questions:**

The paper is well-presented and well-organized. However, the literature review is suggested to be enriched. The existing studies are hardly investigated. Online learning literature not based on deep learning needs to be further studied, there is a lack of state-of-the-art references. Moreover, cross-variable dependency has already been studied in online learning as well as ensemble learning based methods.

Cross-variable dependency reference:

“Minimax classification under concept drift with multidimensional adaptation and performance guarantees”

Ensemble learning references:

“Online Ensemble Multi-kernel Learning Adaptive to Non-stationary and Adversarial Environments”

“Dynamic Weighted Majority: An Ensemble Method for Drifting Concepts”

“Incremental Learning of Concept Drift in Nonstationary Environments”

I think one improvement the authors can consider is to add some discussions about the computational and memory complexity. In online learning, we sometimes need to obtain real-time predictions.

Comparing the theoretical results with similar results from the state-of-the-art would help to show the contributions of the presented bounds.

References:

“Online Ensemble Multi-kernel Learning Adaptive to Non-stationary and Adversarial Environments”

“An adaptive gradient method for online AUC maximization”

“Large Scale Online Kernel Learning”

The reviewer suggests making the code available so that the results can be replicated.


**Limitations:**

.

---

> ### Author Rebuttal · Authors · 2023-08-10
>
> We sincerely apologize for any confusion caused by the lack of clarity in our paper. We appreciate your feedback and would like to address your comments to provide a clearer understanding of our work.
>
> ----
>
> **Concern 1 Online learning literature not based on deep learning**
>
> Regarding the literature review, we acknowledge your suggestion and will enrich our work with more references from online learning not based on deep learning. However, we would like to clarify that our paper's main focus is on online time series forecasting, which is a specific and challenging problem (**See general response for detailed discussion**).
>
> As the primary goal of this paper is to focus on model architecture and empirical performance, the theoretical results only serve to illustrate the rationality of our design. Although in the current draft, the order of theoretical bounds is similar to the standard result in the literature, it is sufficient to justify the design philosophy behind our proposed components: two-stream forecasters and decoupled training strategy. Due to the scope of this paper, we will hold the in-depth theoretical analysis for future research.
>
>
> ----
>
> **Concern 2 cross-variable dependency has already been studied in online learning**
>
> Thank you for pointing out the reference [1], which is a statistical learning method designed for supervised classification, and it requires instance-label pairs, making it fundamentally different from our task of time series forecasting.
>
> In our context, when we mention cross-variable dependency, we refer to the correlation and interactions among different input variables within a time series forecasting problem, which is not the same as [1]. We acknowledge the importance of clarifying the differences between the terminologies used in different contexts. In the revised version of the paper, we will provide more explicit definitions and explanations to distinguish our usage of cross-variable dependency from that of [1] and ensure a better understanding of the specific challenges and solutions in time series forecasting with our proposed OneNet approach.
>
> [1] V. Alvarez, et al., Minimax classification under concept drift with multidimensional adaptation and performance guarantees, ICML 2022.
>
> ----
>
> **Concern 3 ensemble learning based methods**
>
> First and foremost, we would like to emphasize that our method is not a simple ensemble approach. Our first core contribution lies in **recognizing the limitations of a single model in handling concept drift and devising a way to train two complementary prediction models**. This motivates us to explore ensemble learning algorithms. As shown in Table 4, we find that by using the two complementary models we proposed, even with the simplest ensemble strategy (averaging), significant improvements can be achieved compared to the SOTA method.
>
> Moreover, we introduce advanced ensemble learning baselines, such as Mixture-of-experts [1] and Gating mechanism [2], which are more relevant and real-time in the context of our work. As for the baselines you mentioned, [4] is suitable for classification tasks as it requires judging the correctness of the classification results to update expert weights and create/delete experts. The voting strategy in [3.5] can be adapted for our task with some modifications. We implement it and report the results in the Table.1 of the attached pdf.
>
> As shown in the table, the method designed for online multiple kernel learning [3] performs poorly in this setting and is even outperformed by our simple baseline (averaging the prediction results of two forecasters). The strategy in [5] yields slightly better results than simple averaging, but in our scenario, it still falls short compared to our ensemble baseline (Gating) and especially our proposed OneNet method.
>
> [1] Outrageously large neural networks: The sparsely-gated mixture-of-experts layer.
>
> [2] Gated transformer networks for multivariate time series classification
>
> [3] Online ensemble multi-kernel learning adaptive to non-stationary and adversarial environments
>
> [4]  Dynamic weighted majority: An ensemble method for drifting concepts
>
> [5] Incremental learning of concept drift in nonstationary environments
>
> ----
>
> **Concern 4 computational and memory complexity**
>
> We understand the importance of computational and memory complexity in online learning, and we agree that discussing these aspects in our paper would be beneficial. Using ensemble techniques would naturally increase the model size and training time. The statistics are shown in the table below, and it can be observed that our method (OneNet) does not sacrifice too much in terms of inference time compared to previous SOTA solution [1]. However, it is inevitable that there will be some memory overhead. (Table.2 in the attached pdf)
>
> Nevertheless, it is worth mentioning that in this paper, we have explored ways to mitigate this issue. Starting from line 278, we propose a parameter-efficient version called OneNet-, which significantly reduces the number of parameters and speeds up the inference process compared to the traditional framework. Moreover, its performance is only slightly lower than OneNet. This allows us to strike a better balance between computational complexity and forecasting accuracy.
>
> [1]  Learning fast and slow for online time series forecasting, ICLR, 2023
>
> ----
>
> **Concern 5 Comparing the theoretical results with similar results**
>
> According to your suggestion, in our revised version, we will include a separate subsection in the related work to discuss relevant online learning theory works more comprehensively.
>
> ----
>
> **Concern 6 making the code available**
>
> We have uploaded the Anonymous GitHub link.
>
> https://anonymous.4open.science/r/OneNet-58B3
>
> ----
>
> Once again, we thank you for your valuable feedback, and we will take your suggestions into account while revising the paper. If you have any further questions or need more information, please feel free to let us know.

---

> > ### Author Response · Authors · 2023-08-16
> > **Author-Reviewer discussion phase ends on Aug 21st 1pm EDT**
> >
> > We hope this message finds you well. As the deadline for the review and feedback discussion process is rapidly approaching, we would like to kindly remind you of the urgency of your response to our recent comments. Your valuable insights and feedback play a crucial role in shaping the quality and rigor of our work.
> >
> > Additionally, we've noticed that Reviewer r9Pc has raised a similar question to yours. In our ongoing discussion with Reviewer r9Pc, we have addressed several concerns and provided comprehensive clarifications that might aid in your understanding of our contribution as well.
> >
> > Given the limited time remaining, we sincerely request your prompt engagement in the review process. Your perspective is invaluable to us, and we are eager to ensure that all concerns and questions are addressed before the review period concludes.
> >
> > Thank you for your time and consideration. We eagerly await your response and look forward to the opportunity to further enhance the clarity and depth of our work based on your valuable feedback.

---

### Official Review · Reviewer_QKSm · 2023-07-02

**Soundness:** 3 good
**Presentation:** 4 excellent
**Contribution:** 3 good
**Rating:** 7
**Confidence:** 3

**Summary:**

Motivated by the observation that a single model may not consistently achieve optimal performance in time series forecasting, the author proposes a method, which combines two stream architectures (cross-time and cross-variable models) using online convex programming. However, due to the limitations of online convex programming, which involves a trade-off between switching speed and overall performance, an RL-based algorithm is introduced as a solution. The proposed method is reinforced by a range of experimental results, providing support for its effectiveness.

**Strengths:**

1. Easy to follow:
The paper is excellently written and presented in a manner that is easy to comprehend and follow along.

2. Novelty:
While ensemble methods for time series tasks have been previously proposed, the specific approach taken in this paper, such as the adoption of OCP and RL for model ensembling, is novel and has not been introduced before. This novelty serves as a notable advantage of this paper.

3. Concrete experimental results:
The paper provides a comprehensive comparison between its proposed method and numerous baselines across various dataset settings. Additionally, to assess the effectiveness of their ensembling approach, the paper includes comparisons with multiple baseline ensembling methods. These extensive and well-designed experimental results serve to reinforce the credibility and robustness of the paper.

**Weaknesses:**

There are minor issues in this paper.

1. About the motivation:
In the introduction section, the author presents Figure 1 and Table 1 as part of the motivation for their work. The main motivation behind this study is the recognition that a single model is insufficient to address complex time series problems, leading to the proposal of an ensemble method. However, the necessity of Table 1 to explain this motivation is not clear. Regarding Table 1, the author mentions that "cross-time forecasters tend to exhibit lower performance in certain datasets, and existing models that leverage both cross-variable and cross-time information perform worse than the naive TCN method." This sentence may not be directly relevant to the motivation, and it would be beneficial to rephrase it to establish a closer connection between the sentence and the motivation.

2. Ablation Study:
It would be valuable to include experimental results from ablating the offline RL for short-term weight. This ablation study can provide concrete evidence for the necessity of this module in a more tangible manner.

**Questions:**

In addition to the question raised in the weaknesses section, I have another question.

Could you provide a more detailed algorithmic description that illustrates the training or inference process using the \usepackage{algorithm} package? This would greatly aid in understanding the intricacies of the algorithm.

**Limitations:**

The paper gives good motivation, novel methods, and concrete experimental results. I think this paper deserves acceptance.
Addressing these concerns can further enhance the overall quality and strength of the paper. If some minor concerns which posed in weaknesses and questions section, I will raise my points.

---

> ### Author Rebuttal · Authors · 2023-08-10
>
> Thank you for your valuable feedback and positive evaluation of our paper. We appreciate your thorough review and are glad that you find the paper easy to follow and well-presented.
>
> ----
>
> **Concern 1 About the motivation**
>
> Regarding the motivation, we understand your concern about the clarity of Table 1 in establishing the motivation. We would like to emphasize that the statement "cross-time forecasters tend to exhibit lower performance in certain datasets, and existing models that leverage both cross-variable and cross-time information perform worse than the naive TCN method" aims to highlight specific limitations of existing methods when dealing with complex time series forecasting problems.
>
> Cross-time forecasters, due to their lack of modeling inter-variable relationships, may **exhibit lower performance on datasets with a small number of variables**. On the other hand, the models that heavily leverage cross-variable information (cross-variable forecasters) may easily **overfit to the current pattern in training data when dealing with datasets with a large number of variables**, leading to poor generalization in the presence of concept drift, as particularly observed in the ECL dataset.
>
> Although some existing methods claim to capture both cross-variable and cross-time dependencies, we find that the performance under concept drift is even worse than the naive TCN method. This observation serves as a catalyst for us to propose a new framework, OneNet, which effectively balances the cross-variable and cross-time dependencies under concept drift scenarios, leading to improved generalization performance.
>
>
> We hope this clarification provides a more explicit connection between Table 1 and the motivation behind our work. We will ensure that the revised version of the paper clearly establishes the rationale for introducing OneNet to address the limitations of existing methods in handling complex time series problems with concept drift. If you have any further questions or suggestions, please feel free to let us know.
>
> ----
>
> **Concern 2 Ablation Study**
>
> We did conduct a comprehensive ablation study on the OCP block in Table 4. Specifically, the Exponentiated Gradient Descent (EGD) serves as the baseline without offline RL for short-term weight, while the Reinforcement Learning to learn the weight directly (RL-W) serves as the baseline without long-term weight. From the table, we can observe that solely using short-term weight already yields better results compared to EGD. Additionally, combining both short-term and long-term weight further enhances the performance of our algorithm.
>
> The results from this ablation study indeed provide concrete evidence of the necessity of incorporating both short-term and long-term weights using reinforcement learning in the OneNet framework. It demonstrates that the RL-based algorithm plays a crucial role in achieving improved forecasting performance, especially under concept drift scenarios.
>
> ----
>
> **Concern 3 a more detailed algorithmic description**
>
> Thank you for your suggestion. We have included a more detailed algorithmic description in the uploaded pdf file. This algorithmic description provides a step-by-step illustration of the training and inference process of the OneNet method using the \usepackage{algorithm} package. We believe that this additional information will aid in understanding the intricacies of our algorithm and enhance the clarity of our paper.
>
> ----
>
> If you have any further suggestions or comments, we welcome them and will incorporate any necessary improvements to enhance the overall quality and reliability of the paper. Your feedback is valuable to us, and we are committed to producing a high-quality and impactful contribution.

---

> > ### Comment · Reviewer_QKSm · 2023-08-13
> > **Thank you for answering my questions**
> >
> > I'm pleased to acknowledge the thoroughness of your responses in addressing my concerns. Your explanations have successfully resolved all the questions I had. As such, I'd like to raise my point as I said.
> >
> > **Suggestion**
> >
> > In reference to concern 2, I would like to offer a suggestion. In Table 4, you currently group EGD, RL-W, and other ensemble methods together under the label "Ensembling methods." However, creating distinct categories for EGD and RL-W might enhance the clarity of the presentation. By assigning separate groups for EGD and RL-W, you can provide a clearer and more direct illustration of why both methods are necessary, rather than using either of them. This adjustment could contribute to a more straightforward understanding of the rationale behind using both EGD and RL-W.

---

> > > ### Author Response · Authors · 2023-08-14
> > > **Response to reviewer QKSm**
> > >
> > > Thank you for your positive feedback and for acknowledging the thoroughness of our responses in addressing your concerns. We appreciate your engagement and value your input. Your suggestion to create distinct categories for EGD and RL-W in Table 4 is well-taken and we will definitely assign separate groups for EGD and RL-W for a more clear comparison.

---

### Official Review · Reviewer_7dRB · 2023-07-07

**Soundness:** 3 good
**Presentation:** 3 good
**Contribution:** 3 good
**Rating:** 7
**Confidence:** 3

**Summary:**

This paper addresses the problem of concept drift in online Time Separate forecasting. A method OneNet is proposed that uses temporal correlations and inter-variables dependencies. OneNet is compared with other state of the art methods and OneNet is demonstrated to be superior. Furthermore, through empirical studies various factors are studies that effect the forecasting


**Strengths:**

1. Mathematicsl formulation of the problems and OneNet solution
2. Training of 2 independent forecasts: one for tempra and second for the inter-variables dependencies.
3. The use of long term historical and short term variations
4. Ablation study and analysis
5. Comparison with state of art methods


**Weaknesses:**

Does not describe the datasets used for the experiments, just started using abbreviation in tables.

**Questions:**

Will it be possible to test the proposed OneNet using generated data, so that all the benefits can be highlighted with controlled experiments

**Limitations:**

No Limitations are listed

---

> ### Author Rebuttal · Authors · 2023-08-10
>
> Thank you for your feedback and positive comments. We appreciate your thorough evaluation on our paper. We are glad that you find the mathematical formulation, the training of two independent forecasts, and the use of long-term historical and short-term variations as strengths of our work.
>
> We agree with your observation about the lack of description of the datasets used in the experiments. In the revised version of the paper, we will ensure to first provide simple information about the datasets, including their sources and characteristics.
>
> Regarding your question, yes, it is indeed possible to test the proposed OneNet using synthesized data. However, generating meaningful and realistic multivariate simulation data can be hard. Besides, **controlling the data preference to cross-channel bias and cross-time bias is more challenging**. To ensure that the data used in our experiments is representative and practical, we primarily chose real-world datasets to validate our algorithm. These datasets encompass different numbers of variables (from 7 to 321) and varying degrees of concept drift, and OneNet consistently demonstrates **remarkable performance improvements** across various scenarios.
>
> We will consider conducting meaningful synthetic datasets in future work to further demonstrate the power of OneNet. Your feedback is quite valuable, and we are committed to addressing your comments to enhance the quality and clarity of our work. Thank you for your valuable comment.

---

> ### Comment · Reviewer_7dRB · 2023-08-18
>
> I am thankful to the Author for answering my questions. Although I still think using some synthetic data will greatly enhance the value and novelty of the paper.  In any case, I encourage the Authors to publish extended version of this work in a Journal after validating their approach using synthetic data.
>
> Overall, I am satisfied with my original rating.

---

### Official Review · Reviewer_r9Pc · 2023-07-07

**Soundness:** 3 good
**Presentation:** 2 fair
**Contribution:** 2 fair
**Rating:** 5
**Confidence:** 4

**Summary:**

The paper introduces an approach called Online ensembling Network (OneNet), aimed at online updating of time series forecasting models. This approach is crafted to handle the concept drift problem efficiently using streaming data. The authors note that while some algorithms designed for online time series forecasting exploit cross-variable dependency, others presume independence among variables. Each assumption carries its own strengths and weaknesses when it comes to online time series modeling. To address this, the authors' OneNet dynamically updates and combines two models. One model emphasizes on modeling the dependency across the time dimension, while the other is concentrated on cross-variate dependency. The OneNet method infuses a reinforcement learning-based strategy into the conventional online convex programming framework. This integration allows for the linear combination of the two models with dynamically adjusted weights. By doing so, OneNet addresses a significant drawback of classic online learning methods, which often struggle to adapt swiftly to the concept drift. Empirical findings showcase that OneNet diminishes online forecasting error by over 50% compared to the State-Of-The-Art (SOTA) method.






**Strengths:**

1. This paper is easy to follow and read. The considered problem is important in the real world.

2. The proposed method works well in many experiments, verifying its effectiveness. The experiments are solid.

**Weaknesses:**

1. This paper's technical novelty is limited. The authors do not analyse why their method works in theory. Updating techniques itself is not novel, thus theoretical analysis is necessary for this paper, showing the proposed method should work well and work better than other methods in theory. The proposed theory of this paper is not specific and the corresponding bound is not tight and is less informative.

2. In the introduction, I do not see a relevant discussion regarding the theory of online learning.

3. In time series, there will be dependence among data. Will this kind of dependency influence the generalization result? Under what kind of assumption, it will not influence the generalization result?


**Questions:**

see above.

---

> ### Author Rebuttal · Authors · 2023-08-10
>
> Thank you for your valuable feedback and comments on our paper. We appreciate the positive feedback on the readability and the importance of the considered problem in the real world. We also acknowledge the points you raised regarding the technical novelty and the need for theoretical analysis in the paper.
>
> **Concern 1: this paper's technical novelty**
>
> In deep learning domain, developing a generalization bound under changing environments for forecasting tasks is indeed a challenging task but also beyond the primary focus of this paper. We will take your suggestion into account and work towards providing a more comprehensive theoretical analysis, including tighter bounds and more informative insights into the generalization results of the proposed OneNet approach.
>
> However, note that **our work is primarily empirical in nature**, with a focus on building a stronger online time series forecaster. The theoretical framework presented in the manuscript is employed not as the central theme, but rather as a means to substantiate the rationality of our proposed methodology and to elucidate the motivation behind it.
>
> Furthermore, the previous work closest to ours should be [1], which also extensively discusses the gap between online learning and our specific setting. In our experiments, we have also included various online learning baselines for comparison. We find that traditional online learning algorithms do not perform exceptionally well in this setting. This highlights the need for designing algorithms or models that are better suited for this specific issue of concept drift. We believe that our empirical validation is a valuable contribution to the field of robust time series forecasting.
>
> Additionally, we conclude the technical novelties as follows for your convenience:
>
> 1. As far as we know, this is the first work that evaluates almost all advanced structures/design choices of time series forecasting under concept drift settings. Our experiments shed light on the architectural design of robust forecasting models.
> 2. We observe the data preference for cross-variable dependency and cross-time biased models and point out why a single model cannot perform well under all concept-drift settings. Thus we propose OneNet to utilize the strengths of both kinds of models.
> 3. OneNet addresses a significant drawback of classic online learning methods and allows for the linear combination of the two kinds of models with data-driven adjusted weights. OneNet reduces the average cumulative mean-squared errors (MSE) by 53.1% and mean-absolute errors (MAE) by 34.5%. In particular, the performance gain on the challenging dataset ECL is superior, where the MSE is reduced by 59.2% and MAE is reduced by 63.0% compared to the SOTA model FSNet[1].
>
>
> [1] Pham, Quang, et al. "Learning fast and slow for online time series forecasting." ICLR , 2023
>
>
> ----
>
> **Concern 2: a relevant discussion regarding the theory of online learning**
>
> Our focus is online time series forecasting, which is mainly for real-world applications, therefore, the theoretical analysis in this paper is just trying to explain the motivation of OneNet and make the design choice easy to understand. So we just introduce some basic online learning theory works in section 3. According to your suggestion, in our revised version, we will include a separate subsection in the related work to discuss relevant online learning theory works more comprehensively. This will help readers to have a deeper understanding of the theoretical foundations and the relationship between OneNet and existing online learning approaches.
>
> ----
>
> **Concern 3: will dependence among data influence the generalization result?**
>
> Your question may encompass two distinct forms of data dependence: cross-variable dependency and cross-time dependency. However, we would appreciate further clarification on the specific type of dependence you are referring to, as the implications for generalization may differ based on the context.
>
> **Cross-variable dependency:** In time series data, cross-variable dependencies widely exist, meaning that different variables in time series may be correlated or influenced by each other over time. However, it is not accurate to say that cross-variable dependency directly affects generalization ability. The impact on generalization ability depends more on the model and the presence of distribution drift. For example, in usage settings with a large number of variables (ECL dataset in Table.1), the models that heavily focus on cross-variable dependency may overfit the training instances, leading to poor generalization performance when facing distribution drift in a changing environment.  If there is no significant distribution shift between the training and test sets, the impact of cross-variable dependency on generalization ability will not be pronounced.
>
> **Cross-time dependency:** Another type of dependence among data would be cross-time dependency. In literature, people usually assume a stationary beta-mixing autoregressive process to control the time series’s “volatility” or “variance” and derive the generalization error bound. (e.g., [1] and [2]).  However, in our setting, the underlying model evolves over time, i.e., a non-stationary autoregressive process, which makes the analysis even harder. We totally agree with reviewer that a rigorous generalization bound would be very important. Due the scope of this work, we would like to hold it for future research.
>
> [1] McDonald, Daniel J., Cosma Rohilla Shalizi, and Mark Schervish. "Generalization error bounds for stationary autoregressive models." arXiv preprint arXiv:1103.0942 (2011).
>
> [2] Vankadara, Leena Chennuru, et al. "Causal forecasting: generalization bounds for autoregressive models." Uncertainty in Artificial Intelligence. PMLR, 2022.
>
> ----
>
> If you have any further suggestions or comments, we welcome them and will incorporate any necessary improvements.

---

> > ### Comment · Reviewer_r9Pc · 2023-08-14
> >
> > Thanks for the replies.
> >
> > As is claimed, this work is primarily empirical in nature. Therefore, I will not discuss the theoretical framework at this stage.
> >
> > One of the main contributions of this paper is summarised as the first work that evaluates almost all advanced structures/design choices of time series forecasting under concept drift settings. Our experiments shed light on the architectural design of robust forecasting models. Here, I consider the "robust forecasting models" as an online time series forecasting model that can adapt to concept drift. Please correct me if I misunderstand any.
> >
> > If it is to build a robust forecasting model, most concept drift adaptation models designed for continuously labeled data can be a "robust forecasting model". I think reviewer UB5d has listed some related references. Online ensemble model, either model selection-based or model weighting-based, has been widely studied in the concept drift field (note that [1] accurately publishes 10 years ago, listing a collection of existing works suggests use ensemble model rather than a single model) Therefore, I didn't see any novelty in your proposed architectural design if compared to existing concept drift adaptation methodologies. For me, it is merely a simple use of one existing drift adaptation framework.
> >
> > There are no high-level insights into why using the proposed online ensemble architecture perfectly fits your time series forecasting tasks. Considering this is a submission to Neurips, I don't think this paper provides me with a deep understanding of the proposed problem.
> >
> >
> >
> > Another contribution of "OneNet addresses a significant drawback of classic online learning methods and allows for the linear combination of the two kinds of models with data-driven adjusted weights." Here, I understand the significant drawback of classic online learning methods as "the main shortcoming of classical online learning methods that tend to be slow in adapting to the concept drift"
> >
> > Firstly, your baseline methods are not designed to handle concept drift. Certainly, their performance could be low when applied to data with drift. Therefore, the experiments are not fair from my perspective. In addition, drift can occur in different types. It is not always good to fast adapt to new data. For me, there should be at least a more careful consideration of your defined shortcoming of classical online learning.
> >
> > Overall, the proposed method is considered under concept drift. The presented idea shows some potential. However, the current edition does need major improvements on existing concept drift methodologies from my perspective.

---

> > > ### Author Response · Authors · 2023-08-14
> > > **Response [1/3]**
> > >
> > > We appreciate your discerning review, which drives us to enhance the clarity and depth of our study.
> > >
> > > ----
> > >
> > > **Concern 1.1 Online ensemble model, either model selection-based or model weighting-based, has been widely studied in the concept drift field**
> > >
> > > **Concern 1.2 why using the proposed online ensemble architecture perfectly fits time series forecasting tasks**
> > >
> > > First and foremost, we would like to emphasize that our method is not a simple ensemble approach. Our first core contribution lies in *pointing out the data preference for cross-variable dependency and cross-time biased models* and explaining why a single model cannot perform well under all concept-drift settings. This motivates us to explore ensemble learning algorithms and is also the reason for **why using the proposed online ensemble architecture perfectly fits time series forecasting tasks**. As shown in Table 4, we find that by using the two complementary models we proposed (cross-variable dependency and cross-time biased models), even with the simplest ensemble strategy (averaging), significant improvements can be achieved compared to the SOTA method. Namely, the ensembling strategy is a small part of our method and the main performance gains are from the complementary model selection.
> > >
> > > Moreover, we introduce advanced ensemble learning baselines, such as Mixture-of-experts [1] and Gating mechanism [2]. As for the baselines that reviewer UB5d mentioned, [4] is suitable for classification tasks as it requires judging the correctness of the classification results to update expert weights and create/delete experts. The voting strategy in [3, 5] can be adapted for our task with some modifications. We implement it and report the results as follows:
> > >
> > >
> > > |                       |       Method      | ETTH2 |       |       |  ECL  |       |       |  Avg  |
> > > |:---------------------:|:-----------------:|:-----:|:-----:|:-----:|:-----:|:-----:|:-----:|:-----:|
> > > |                       |                   |   1   |   24  |   48  |   1   |   24  |   48  |       |
> > > |        Baseline       |       FSNet       | 0.466 | 0.687 | 0.846 | 3.143 | 6.051 | 7.034 | 3.038 |
> > > | Ensembling  baselines |      Average      | 0.381 | 0.607 | 0.595 | 2.458 | 2.833 | 3.309 | 1.697 |
> > > |                       |     Gating [2]    | 0.476 | 0.678 | 0.782 | 2.474 | 2.181 | 2.301 | 1.482 |
> > > |                       |      MOE [1]      | 0.488 | 0.565 | 1.238 | 3.312 | 3.086 | 2.497 | 1.864 |
> > > |                       |    AdaRaker[3]    | 0.489 | 0.723 | 0.934 | 2.742 | 2.796 | 2.842 | 1.754 |
> > > |                       | Learn ++. NSE [5] | 0.383 | 0.667 | 0.672 | 2.623 | 2.324 | 2.412 | 1.514 |
> > > |          Ours         |       OneNet      | 0.380 | 0.532 | 0.609 | 2.351 | 2.074 | 2.201 | 1.358 |
> > >
> > > As shown in the table, the method designed for online multiple kernel learning [3] performs poorly in this setting and is even outperformed by our simple baseline (averaging the prediction results of two forecasters). The strategy in [5] yields slightly better results than simple averaging, but in our scenario, it still falls short compared to our ensemble baseline (Gating) and especially our proposed OneNet method.
> > >
> > >
> > > [1] Shazeer, Noam, et al. "Outrageously large neural networks: The sparsely-gated mixture-of-experts layer." arXiv preprint arXiv:1701.06538 (2017).
> > >
> > > [2] Liu, Minghao, et al. "Gated transformer networks for multivariate time series classification." arXiv preprint arXiv:2103.14438 (2021).
> > >
> > > [3] Shen, Y., Chen, T., & Giannakis, G. (2018, March). Online ensemble multi-kernel learning adaptive to non-stationary and adversarial environments. In International Conference on Artificial Intelligence and Statistics (pp. 2037-2046). PMLR.
> > >
> > > [4]  Kolter, J. Z., & Maloof, M. A. (2007). Dynamic weighted majority: An ensemble method for drifting concepts. The Journal of Machine Learning Research, 8, 2755-2790.
> > >
> > > [5] Elwell, R., & Polikar, R. (2011). Incremental learning of concept drift in nonstationary environments. IEEE Transactions on Neural Networks, 22(10), 1517-1531.

---

> > > > ### Author Response · Authors · 2023-08-14
> > > > **Response [2/3]**
> > > >
> > > > **Concern 2 I didn't see any novelty in your proposed architectural design if compared to existing concept drift adaptation methodologies.
> > > > We appreciate your thoughtful evaluation and the attention you've given to our proposed architectural design.**
> > > >
> > > > While the concept drift adaptation methods have been discussed in various literature, how to do it effectively for time series forecasting remains an open question. We would like to highlight some key designs that contribute to the novelty and effectiveness.
> > > >
> > > > **Complementary Model Selection:** OneNet's strength lies in the careful selection of two distinct models—cross-time and cross-variable—that complement each other's strengths. This strategic combination allows us to capture a wider range of patterns and dynamics in the data, enabling more accurate and robust forecasting.
> > > >
> > > > **Long-term and Short-term Weights:** We acknowledge the importance of addressing the limitations of traditional Online Convex Programming (OCP) frameworks. In OneNet, we address these flaws by introducing an Offline reinforcement learning framework to learn short-term weights. By incorporating short-term weight learning, we empower our model to capture rapid changes in data patterns, further enhancing its forecasting accuracy.
> > > >
> > > > While OneNet's foundation builds upon established concepts, its unique combination of model selection, short-term weight learning, and comprehensive experimental validation sets it apart from a mere replication of existing methodologies.  OneNet reduces the average cumulative mean-squared errors (MSE) by **53.1%** and mean-absolute errors (MAE) by **34.5%** compared to the SOTA models, which is of great value to real-world development.
> > > >
> > > > ----
> > > >
> > > > **Concern 3 your baseline methods are not designed to handle concept drift**
> > > >
> > > > It's important to acknowledge that the landscape of concept drift baselines may not be fully extensive, and the utilization of online learning for handling concept drift in time series data remains an area that is still emerging. While online learning methods have been extensively studied, their application in online time series forecasting poses an ongoing challenge that warrants further exploration.
> > > >
> > > > In this paper, we have strived to comprehensively address the range of possible baselines, categorizing them into four distinct groups:
> > > >
> > > > **Time Series Forecasting Models**: We have presented results for several time series forecasting baseline methods in Tables 2, 3, and 12, including OnlineTCN, Informer, and TS-Mixer. Notably, these models exhibit lower performance under the presence of concept drift, confirming your observation that such naive fine-tuning approaches may not effectively handle concept drift.
> > > >
> > > > **Online Learning Baselines**: Our exploration includes online learning baselines such as ER, MIR, and DER++, which showcase notably enhanced performance when dealing with concept drift. These methods are explicitly designed to tackle the complexities posed by evolving data distributions. Our experiments highlight their adaptability to changing patterns, underscoring their relevance in concept drift scenarios.
> > > >
> > > > **Ensembling Learning Baselines**: We've considered ensembling learning baselines, incorporating methodologies like Mixture-of-experts, gating mechanisms, traditional online convex programming, and other ensembling techniques adaptable to continuously labeled data, as suggested by reviewer UB5d.
> > > >
> > > > **FSNet - State-of-the-Art Model:** A prominent addition to our investigation is FSNet, a state-of-the-art model for online time series forecasting, recently introduced in ICLR 2023. Notably engineered to address concept drift, FSNet leverages advanced architecture and mechanisms to excel in contexts where shifts in data distribution are prevalent.

---

> > > > > ### Author Response · Authors · 2023-08-14
> > > > > **Response [3/3]**
> > > > >
> > > > > **Concern 4 It is not always good to fast adapt to new data.**
> > > > >
> > > > > Regarding your observation that "It is not always good to fast adapt to new data," we fully acknowledge the complexity of concept drift scenarios and the potential risks associated with rapid adaptation. While we concur with your viewpoint, we would like to clarify that our experiments do not provide explicit evidence of harmful effects from adapting to new data for the baseline methods. The following table provides the performance of some advanced forecasting models and their online finetuning counterparts under concept drift.
> > > > >
> > > > > |                    |         | ETTH2  |        |        | ECL     |         |         |         |
> > > > > |--------------------|---------|--------|--------|--------|---------|---------|---------|---------|
> > > > > | Method             | offline |      1 |     24 |     48 |       1 |      24 |      48 | Avg     |
> > > > > | CrossFormer        | √       | 23.270 | 28.904 | 29.218 | 469.260 | 475.490 | 478.270 | 250.735 |
> > > > > |                    | ×       |  9.873 |  2.856 |  5.772 |  68.300 |  92.500 |  94.790 |  45.682 |
> > > > > | Fedformer-Wavelets | √       |  1.816 |  3.070 |  3.996 |  41.791 |  37.236 |  37.210 |  20.853 |
> > > > > |                    | ×       |  1.798 |  2.993 |  1.623 |  21.387 |  24.600 |  27.640 |  13.340 |
> > > > > | TCN                | √       | 27.060 | 27.760 | 26.320 | 538.000 | 546.000 | 552.000 | 286.190 |
> > > > > |                    | ×       |  0.530 |  0.930 |  0.910 |   3.010 |  11.680 |  10.800 |   4.643 |
> > > > > | Time-TCN           | √       |  4.530 |  7.840 |  1.300 |  47.900 |  48.660 |  67.150 |  29.563 |
> > > > > |                    | ×       |  0.480 |  0.780 |  1.300 |   4.010 |   5.220 |   5.210 |   2.833 |
> > > > > | DLinear            | √       |  2.910 | 10.250 |  7.530 |  12.030 |  51.280 |  58.460 |  23.743 |
> > > > > |                    | ×       |  2.440 |  9.240 |  6.910 |   6.690 |  27.820 |  31.540 |  14.107 |
> > > > > | TS-Mixer           | √       |  1.968 |  3.525 |  4.880 |  11.160 |  30.930 |  44.680 |  16.191 |
> > > > > |                    | ×       |  0.780 |  2.050 |  3.060 |   2.798 |   4.983 |   5.764 |   3.239 |
> > > > >
> > > > >
> > > > > In our empirical evaluations, we have consistently observed that finetuning the baseline models with new data leads to notably improved performance compared to their offline counterparts. This suggests that, within the context of our experiments, adapting to new data is indeed beneficial and aligns with the primary objective of online forecasting methods—to adapt and generalize to changing patterns.
> > > > >
> > > > > ----
> > > > >
> > > > > Again, thank you for recognizing the potential of our proposed method within the concept drift context. We will take your feedback into account as we strive to make significant improvements to our existing methodologies. Your insights provide valuable guidance as we work towards refining and strengthening our approach.
> > > > >
> > > > > Please feel free to share any additional thoughts or suggestions. We value your expertise and remain dedicated to improving the quality and validity of our research.

---

> > > > > > ### Comment · Reviewer_r9Pc · 2023-08-15
> > > > > >
> > > > > > Thanks for the quick replies.
> > > > > > Thanks for the answers to Concern 1.2 and Concern 4.
> > > > > > "Our first core contribution lies in pointing out the data preference for cross-variable dependency and cross-time biased models and explaining why a single model cannot perform well under all concept-drift settings. "
> > > > > > Here, does your defined "a single model" include the model that ensembles different cross-variable dependency-based models?

---

> > > > > > > ### Author Response · Authors · 2023-08-15
> > > > > > > **Response to reviewer r9Pc**
> > > > > > >
> > > > > > > Certainly, we appreciate your careful consideration of our core contribution and the specific context of our defined "a single model."
> > > > > > >
> > > > > > > In our paper, the reference to "a single model" pertains to either a cross-variable dependency-based model or a cross-time dependency-based model, rather than encompassing an ensemble of different cross-variable dependency-based models. We acknowledge that your inquiry sheds light on the potential interpretation of an ensemble of cross-variable dependency-based models as a collective "single model."
> > > > > > >
> > > > > > > Your suggestion is intriguing, and actually, we have conducted empirical investigations to explore this avenue. However, our findings demonstrated that ensembling various cross-variable dependency-based models did not yield superior performance gains. Thus, ensembling a model group with the same model bias did not fundamentally alter the dynamics or bias of a single cross-variable dependency-based model.
> > > > > > >
> > > > > > > We sincerely value your engagement and your insightful questions, which stimulate our deeper understanding and encourage us to refine our explanations. Please do not hesitate to share further thoughts or queries, as your perspectives contribute significantly to the richness of our discourse.

---

> > > > > > > > ### Comment · Reviewer_r9Pc · 2023-08-15
> > > > > > > >
> > > > > > > > I was asking because I think I might misunderstand your contributions when claiming this ensemble model.
> > > > > > > >
> > > > > > > > According to your replies, it should be an ensemble that combines the temporal dependency and the cross-variate dependency can fast adapt to new data for time series forecasting under concept drift. This would be more reasonable for me to understand this architecture as the data are time series.
> > > > > > > > Honestly, "combining two models is better than one model" as the motivation did misguide me a lot.
> > > > > > > > May I know if code is available for this work?

---

> > > > > > > > > ### Author Response · Authors · 2023-08-15
> > > > > > > > > **Response to reviewer r9Pc**
> > > > > > > > >
> > > > > > > > > We deeply appreciate your feedback and understanding, and we apologize for any confusion our initial presentation may have caused. Your insights have guided us toward clarifying the essence of our contribution.
> > > > > > > > >
> > > > > > > > > Indeed, you are correct. The core motivation and central contribution of our work revolve around the **recognition of the critical significance of combining temporal and cross-variable dependencies**, specifically tailored for the context of time series forecasting under concept drift. This emphasis on the integration of these dependencies, rather than the mere juxtaposition of two disparate models, underscores our approach's efficacy.
> > > > > > > > >
> > > > > > > > > In response to your inquiry, we are pleased to share that the code for our work is available for your reference. You can access it conveniently through the following link:
> > > > > > > > >
> > > > > > > > > https://anonymous.4open.science/r/OneNet-58B3
> > > > > > > > >
> > > > > > > > > We genuinely appreciate your diligence in engaging with our research and your valuable feedback. Please feel free to explore the code repository and do not hesitate to reach out if you have further inquiries or if there are any additional aspects of our work you would like to discuss.

---

### Author Rebuttal · Authors · 2023-08-10

We thank the Reviewers for the insightful comments and detailed feedback. We are delighted that reviewers find our paper has **good motivation** (QKSm), **easy to follow** (QKSm, r9Pc), **excellently -presented and organized** (UB5d, QKSm), **enough technical novelty** (QKSm, 7dRB), and **Impressive empirical results** (r9Pc, QKSm, 7dRB).

There is a shared concern from Reviewers r9Pc and UB5d regarding the novelty of our work in comparison to the theory of traditional online learning literature and some ensembling methods designed for classification tasks or online kernel learning. We would like to emphasize that our work is primarily empirical in nature, with a focus on building a stronger online time series forecaster. *The theoretical framework presented in the manuscript is employed not as the central theme, but rather as a means to substantiate the rationality of our proposed methodology and to elucidate the motivation behind it*. Our key contribution is a novel solution to online time series forecasting inspired by the different data preferences to model biases during the concept drift. It is crucial to note that directly applying common online learning frameworks for image data to time series is not straightforward due to the differences between these two types of data, such as discrete vs. continuous labels and distinct label differences among tasks. Besides, we have included various online learning and ensembling baselines in the main paper and the rebuttal process to address these concerns, hoping that these experiments demonstrate the significance of our proposed approach. We refer the Reviewers to our responses to **Concern #1 of Reviewer r9Pc** for a detailed discussion regarding the novelty of our work.

In the revised version, we have made substantial improvements based on the Reviewers' comments. We include a separate subsection in the related work to discuss relevant online learning theory and online ensembling learning works more comprehensively. Additionally, we provide a more detailed algorithmic description to illustrate the training or inference process, addressing Reviewer QKSm's concern about clarity. Furthermore, we incorporate more ensembling baselines and add discussions about computational and memory complexity to address concerns raised by reviewer UB5d.

---

### Decision · Program_Chairs · 2023-09-21

**Decision:**

Accept (poster)

**Comment:**

This paper deals with multivariate forecasting in the online context, where data streams in, and where concept drift potentially occurs. The authors argue that both dependencies between variables, and along time (independent between variables) are important for good practical performance in this setting. They propose a dynamic combination of two models, each representing one such mode of dependencies. This is not an ensemble, since weights change over time. They incorporate a reinforcement learning algorithm into online convex programming in order to adapt the weights. Their experiments on a number of realistic benchmarks shows a significant improvement over the SotA.

This works addresses an important problem. Reviewers agree the method is novel (esp the use of RL), and the experiments are very well done, the writing is clear. Different to many other online learning papers (also on forecasting), the paper does not present strong theoretical results, but the authors convincingly argue that the main focus of the paper is on good empirical performance in the real world. There are also ablation experiments, showing that RL dynamic weighting is essential for good performance. One possibility of improvement is to include baselines into the comparison which also handle concept drift.